# Uncertainty-Aware Decision Transformer for Stochastic Driving Environments

## Abstract

Offline Reinforcement Learning (RL) has emerged as a promising framework for learning policies without active interactions, making it especially appealing for autonomous driving tasks. Recent successes of Transformers inspire casting offline RL as sequence modeling, which performs well in long-horizon tasks. However, they are overly optimistic in stochastic environments with incorrect assumptions that the same goal can be consistently achieved by identical actions. In this paper, we introduce an **UN**certainty-awa**RE** deci**S**ion **T**ransformer (UNREST) for planning in stochastic driving environments without introducing additional transition or complex generative models. Specifically, UNREST estimates state uncertainties by the conditional mutual information between transitions and returns, and segments sequences accordingly. Discovering the 'uncertainty accumulation' and 'temporal locality' properties of driving environments, UNREST replaces the global returns in decision transformers with less uncertain truncated returns, to learn from true outcomes of agent actions rather than environment transitions. We also dynamically evaluate environmental uncertainty during inference for cautious planning. Extensive experimental results demonstrate UNREST's superior performance in various driving scenarios and the power of our uncertainty estimation strategy.

## 1 Introduction

Safe and efficient motion planning has been recognized as a crucial component and the bottleneck in autonomous driving systems (Yurtsever et al., 2020). Nowadays, learning-based planning algorithms like imitation learning (IL) (Bansal et al., 2018; Zeng et al., 2019) and reinforcement learning (RL) (Chen et al., 2019a; 2020) have gained prominence with the advent of intelligent simulators (Dosovitskiy et al., 2017; Sun et al., 2022b) and large-scale datasets (Caesar et al., 2021). Building on these, offline RL (Diehl et al., 2021; Li et al., 2022a) becomes a promising framework for safety-critical driving tasks to learn policies from offline data while retaining the ability to leverage and improve over data of various quality (Fujimoto et al., 2019; Kumar et al., 2020).

Nevertheless, the application of offline RL approaches still faces practical challenges. Specifically: (1) The driving task requires conducting *long-horizon planning* to avoid shortsighted erroneous decisions (Zhang et al., 2022); (2) The *stochasticity* of environmental objects during driving also demands real-time responses to their actions (Diehl et al., 2021; Villaflor et al., 2022).

The recent success of the Transformer architecture (Vaswani et al., 2017; Brown et al., 2020; Dosovitskiy et al., 2020) has inspired researchers to reformulate offline RL as a sequence modeling problem (Chen et al., 2021), which naturally considers outcomes of multi-step decision-making and has demonstrated efficacy in long-horizon tasks. Typically, they train models to predict an action based on the current state and an outcome such as a desired future return (i.e. reward-to-go). However, existing works (Brandfonbrener et al., 2022; Paster et al., 2022; Yang et al., 2022) have pointed out that these decision transformers (DTs) tend to be overly optimistic in stochastic environments because they incorrectly assume that actions, which successfully achieve a goal once, can consistently do so in subsequent attempts. This assumption is easily broken in stochastic environments, as the goal can be achieved by accidental environment transitions. In Fig. 1(a), identical turning actions may yield entirely different outcomes w.r.t. the aggressive or cautious behavior of the surrounding blue vehicle.

The key to overcoming the problem is to distinguish between outcomes of decisions and environment transitions, and train models to pursue goals that are not affected by environmental stochasticity.

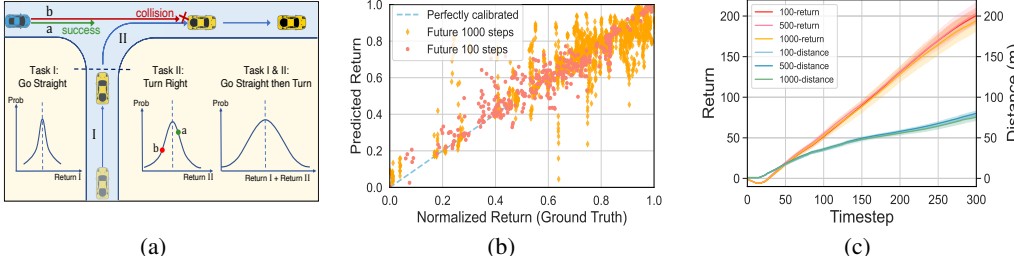

(a)              (b)              (c)

Figure 1: Motivations of UNREST. (a): An example driving scenario where the variance of return increases when accounting for multiple tasks. (b): Calibration results of return distribution over future 1,000 steps are obviously more uncertain than that over future 100 steps. (c): Rollout return and distance curves are close for policies that maximize the return of future 100, 500, and 1,000 steps.

To our best knowledge, only limited works (Villaflor et al., 2022; Paster et al., 2022; Yang et al., 2022) attempt to solve the problem. Generally, they fit a state transition model, either for pessimistic planning (Villaflor et al., 2022) through sampling from VAEs (Zhang et al., 2018), or to disentangle conditioning goals from environmental stochasticity (Paster et al., 2022; Yang et al., 2022) by adversarial training (Shafahi et al., 2019). While adding complexity, these methods are applicable only when state transition functions can be learned adequately, which is often not the case for driving because of the uncertainty brought by complex interactions and partial observability (Sun et al., 2022a; Murphy, 2000). Furthermore, driving trajectories encompass stochastic impact over an excessive number of timesteps. This dilutes the information related to current timestep decision-making in the outcome return and challenges VAE and adversarial training (Wu et al., 2017; Burgess et al., 2018).

In this paper, we take an initial step to customize DTs for stochastic driving environments without introducing transition or complex generative models. Specifically, our insight comes from the two driving tasks in Fig. 1(a): When going straight, the cumulative rewards from Task I & II (termed as the *global return*) contains too much stochastic influence to provide effective supervision. Nonetheless, a viable strategy involves conditioning solely on the truncated return from Task I to mitigate the influence of environmental stochasticity (less stochastic timesteps, lower return variance), which still preserves rewards over sufficient timesteps for the optimization of current actions. Further experiments validate the point and we summarize the following properties of driving environments:

**Property 1 (Uncertainty Accumulation)** *The impact of environmental stochasticity on the return distribution accumulates while considering more timesteps, as validated in Fig. 1(b).*

**Property 2 (Temporal Locality)** *Driving can be divided into independent tasks, where we only need to focus on the current task without considering those much later. Hence, optimizing future returns over a sufficiently long horizon approximates global return optimization, as shown in Fig. 1(c).*

The remaining problem is to specify the span of truncated returns. Specifically, our proposed **UN**certainty-awa**RE** deci**S**ion **T**ransformer (UNREST) quantifies the impacts of uncertainties by the conditional mutual information between transitions and returns, which bypasses the complexity associated with transition and generative modeling. Sequences are then segmented into *certain and uncertain parts* accordingly. With the minimal impact of uncertainty in 'certain parts', we set the conditioning goal as the cumulative reward to the segmented position (with the number of timesteps), which can reflect true outcomes of decisions and be generalized to attain higher rewards. In contrast, in 'uncertain parts' where the environment is highly stochastic, we disregard the erroneous information from returns (with dummy tokens as conditions) and let UNREST imitate expert actions. Dynamic uncertainty evaluation is adopted during inference for cautious planning. **Highlights are:**

- We present UNREST, an uncertainty-aware sequential decision framework to apply offline RL in long-horizon and stochastic driving environments. The code will be public when published.

- Recognizing the properties of driving environments, we propose a novel model-free environmental uncertainty measurement and segment sequences accordingly. Based on these, UNREST bypasses challenging generative training in previous works, by replacing global returns in DTs with less uncertain truncated returns (or dummy tokens) to learn from the true outcomes of agent actions.

- We extensively evaluate UNREST on CARLA (Dosovitskiy et al., 2017), where it consistently outperforms strong baselines (5.2% and 6.5% driving score improvement in seen and unseen scenarios). Additional experiments also prove the efficacy of our uncertainty estimation strategy.

## 2 RELATED WORKS

In this section, we review works about sequence modeling algorithms in RL and uncertainty estimation strategies as the foundation of our work and highlight the differences and contributions of UNREST. More related works on vehicle planning can be found in App. A.1.

**Offline RL as Sequence Modeling:** Despite the potential to learn directly from offline data and the prospects of a higher performance ceiling (than IL) (Fujimoto et al., 2019), the long-horizon and stochastic nature of driving environments still undermine the efficacy of current offline RL applications in autonomous driving tasks (Diehl et al., 2021; Shi et al., 2021; Li et al., 2022a). Encouraged by the success of Transformer (Vaswani et al., 2017) in sequence modeling tasks (Brown et al., 2020; OpenAI, 2023), a recent line of work (Chen et al., 2021; Janner et al., 2021; Furuta et al., 2022; Lee et al., 2022; Hu et al., 2023) adapts Transformer into RL by viewing offline RL as a sequence modeling problem. Typically, Decision Transformer (DT) (Chen et al., 2021) predicts actions by feeding target returns and history sequences, while Trajectory Transformer (TT) (Janner et al., 2021) further exploits the capacity of Transformer through jointly predicting states, actions, and rewards and planning by beam searching. The theoretical basis of these models is revealed in Generalized DT (Furuta et al., 2022): they are trained conditioned on hindsight (e.g. future returns) to reach desired goals. DTs naturally consider outcomes of multi-step decision-making and have demonstrated efficacy in long-horizon tasks (Chen et al., 2021).

However, recent works (Brandfonbrener et al., 2022; Paster et al., 2022; Yang et al., 2022) have pointed out the fundamental problem with this training mechanism. Specifically, in stochastic environments like autonomous driving, certain outcomes can be achieved by accidental environment transitions, hence cannot provide effective action supervision. Notably, this is in fact a more general problem to do with all goal-conditioned behavior cloning and hindsight relabeling algorithms (Eysenbach et al., 2022; Štrupl et al., 2022; Paster et al., 2020), but in this work we specifically focus on solutions to DTs. To tackle this issue, ESPER (Paster et al., 2022) adopts adversarial training (Shafahi et al., 2019) to learn returns disentangled from environmental stochasticity as a condition. Similarly, DoC (Yang et al., 2022) utilizes variational inference (Zhang et al., 2018) to learn a latent representation of the trajectory, which simultaneously minimizes the mutual information (so disentangled) with environment transitions to serve as the conditioning target. Besides, SPLT (Villaflor et al., 2022) leverages a conditional VAE (Zhang et al., 2018) to model the stochastic environment transitions. As the driving environment contains various interactions that make it difficult to model (Sun et al., 2022a), and the long driving trajectory hinders generative training, our study proposes a novel uncertainty estimation strategy to customize DTs without transition or generative models. Besides, different from EDT (Wu et al., 2023) that dynamically adjusts history length to improve stitching ability, UNREST focuses on segmenting sequences and replacing conditions to address the overly optimistic problem.

**Uncertainty Estimation:** Uncertainty estimation plays a pivotal role in reliable AI. One typical method for uncertainty estimation is probabilistic bayesian approximation, either in the form of dropout (Gal & Ghahramani, 2016) or conditional VAEs (Blundell et al., 2015; Dusenberry et al., 2020), which computes the posterior distribution of model parameters. On the contrary, the deep deterministic methods propose to estimate uncertainty by exploiting the implicit feature density (Franchi et al., 2022). Besides, deep ensembles (Lakshminarayanan et al., 2017a) train $K$ neural networks simultaneously, with each module being trained on a different subset of the data, and use the variance of the network outputs as a measure of uncertainty. In this work, we utilize the ensemble approach, which has been widely used in the literature for trajectory optimization (Vlastelica et al., 2021), online and offline RL (Wu et al., 2021; An et al., 2021), to jointly train $K$ variance networks (Kendall & Gal, 2017b) for estimating the uncertainty of returns.

## 3 PRELIMINARY

We introduce preliminary knowledge for UNREST in this section. To keep notations concise, we use subscripts $t$ or numbers for variables at specific timesteps, Greek letter subscripts for parameterized variables, and bold symbols to denote variables spanning multiple timesteps.

### 3.1 ONLINE AND OFFLINE REINFORCEMENT LEARNING

We consider learning in a Markov decision process (MDP) (Puterman, 2014) denoted by the tuple $\mathcal{M} = (\mathcal{S}, \mathcal{A}, P, r, \gamma)$, where $\mathcal{S}$ and $\mathcal{A}$ are the state space and the action space, respectively. Given

states $s, s' \in \mathcal{S}$ and action $a \in \mathcal{A}$, $P(s'|s, a) : \mathcal{S} \times \mathcal{A} \times \mathcal{S} \rightarrow [0, 1]$ is the state transition function and $r(s, a) : \mathcal{S} \times \mathcal{A} \rightarrow \mathbb{R}$ defines the reward function. Besides, $\gamma \in (0, 1]$ is the discount factor. The agent takes action $a$ at state $s$ according to its policy $\pi(a|s) : \mathcal{S} \times \mathcal{A} \rightarrow [0, 1]$. At timestep $t \in [1, T]$, the accumulative discounted reward in the future, named reward-to-go (i.e. return), is $R_t = \sum_{t'=t}^{T} \gamma^{t'-t} r_{t'}$. The goal of *online RL* is to find a policy $\pi$ that maximizes the total expected return: $J = \mathbb{E}_{a_t \sim \pi(\cdot|s_t), s_{t+1} \sim P(\cdot|s_t, a_t)} \left[ \sum_{t=1}^{T} \gamma^{t-1} r(s_t, a_t) \right]$ by learning from the transitions $(s, a, r, s')$ through interacting with the real environment. In contrast, *Offline RL* makes use of a static dataset with $N$ trajectories $\mathcal{D} = \{\tau_i\}_{i=1}^{N}$ collected by certain behavior policy $\pi_b$ to learn a policy $\pi$ that maximizes $J$, thereby avoiding safety issues during online interaction. Here $\tau = \left\{ (s_t, a_t, r_t, s'_t) \right\}_{t=1}^{T}$ is a collected interaction trajectory composed of transitions with horizon $T$.

## 3.2 OFFLINE REINFORCEMENT LEARNING AS SEQUENCE MODELING

Following DTs (Chen et al., 2021), we pose offline RL as a sequence modeling problem where we model the probability of the sequence token $x_t$ conditioned on all tokens prior to it: $p_\theta(x_t|\boldsymbol{x}_{<t})$, where $\boldsymbol{x}_{<t}$ denotes tokens from step 1 to $(t-1)$, like the GPT (Brown et al., 2020) architecture. DTs consider a return-conditioned policy learning setting where the agent at step $t$ receives an environment state $s_t$, and chooses an action $a_t$ conditioned on the future return $R_t = \sum_{t'=t}^{T} r_{t'}$. This leads to the following trajectory representation:

$$\tau = (R_1, s_1, a_1, R_2, s_2, a_2, ..., R_T, s_T, a_T), \tag{1}$$

with the objective to minimize the action prediction loss, i.e. maximize the action log-likelihood:

$$\mathcal{L}_{\text{DT}}(\theta) = \mathbb{E}_{\tau \sim \mathcal{D}} \left[ -\sum_{t=1}^{T} \log p_\theta(a_t|\tau_{<t}, R_t, s_t) \right]. \tag{2}$$

This supervised learning objective is the cause for DTs' limitations in stochastic environments, as it over-optimistically assumes actions in the sequence can reliably achieve the corresponding returns. At inference time, given a prescribed high target return, DTs generate actions autoregressively while receiving new states and rewards to update the history trajectory.

## 4 APPROACH: UNREST

In this section, we begin with an overview of UNREST's insight and architecture, followed by detailed explanations of the composition modules used in the approach.

### 4.1 MODEL OVERVIEW

An overview of the proposed approach UNREST is illustrated in Fig. 2. To address the overly optimistic issue, our key idea is to quantify the impact of environmental uncertainty, and learn to perform aggressively or cautiously at states with different levels of environmental impacts.

To achieve this, we train *return transformers* with different trajectory inputs to identify the impact of environmental uncertainty, obviating the need for complex transition and generative training. The expert sequences are then segmented into certain and uncertain parts w.r.t. estimated uncertainties, each with relabeled conditioning goals to facilitate the *decision transformer* to learn from outcomes of decisions rather than environment transitions. At test time, an *uncertainty predictor* is involved to guide decision-making at different states. In the following, we introduce each module with details.

### 4.2 RETURN TRANSFORMERS FOR UNCERTAINTY ESTIMATION

Instead of conventional uncertainties that reflect variances of distributions (Wu et al., 2021; Kendall & Gal, 2017a), in this paper we estimate the impact of environmental stochasticity, what we really care about for policy learning, as an indirect measure of environmental uncertainty. In particular, we propose to model the impact of the transition $(s_{t-1}, a_{t-1} \rightarrow s_t)$ on return $R_t$ through their conditional mutual information (Seitzer et al., 2021; Duong & Nguyen, 2022), i.e. the divergence between distributions $p(R_t|\tau_{<t}^{\text{ret}})$ and $p(R_t|\tau_{<t}^{\text{ret}}, s_t)$, where $\tau^{\text{ret}} = \{(s_t, a_t)\}_{t=1}^{T}$.

Specifically, two 'return transformers' are trained to approximate return distributions. Initially, states and actions are embedded by linear layers $f_\varphi^s(\cdot)$ and $f_\varphi^a(\cdot)$. The obtained embeddings are then sequentially processed by two transformers $\mathcal{T}_{\varphi_s}(\cdot)$ and $\mathcal{T}_{\varphi_a}(\cdot)$ separately for return prediction:

$$x_{s_t} = f_\varphi^s(s_t), \quad x_{a_t} = f_\varphi^a(a_t),$$
$$\tilde{x}_{s_t} \sim \mathcal{T}_{\varphi_s}(..., x_{a_{t-1}}, x_{s_t}), \quad \tilde{x}_{a_t} \sim \mathcal{T}_{\varphi_a}(..., x_{s_t}, x_{a_t}). \tag{3}$$

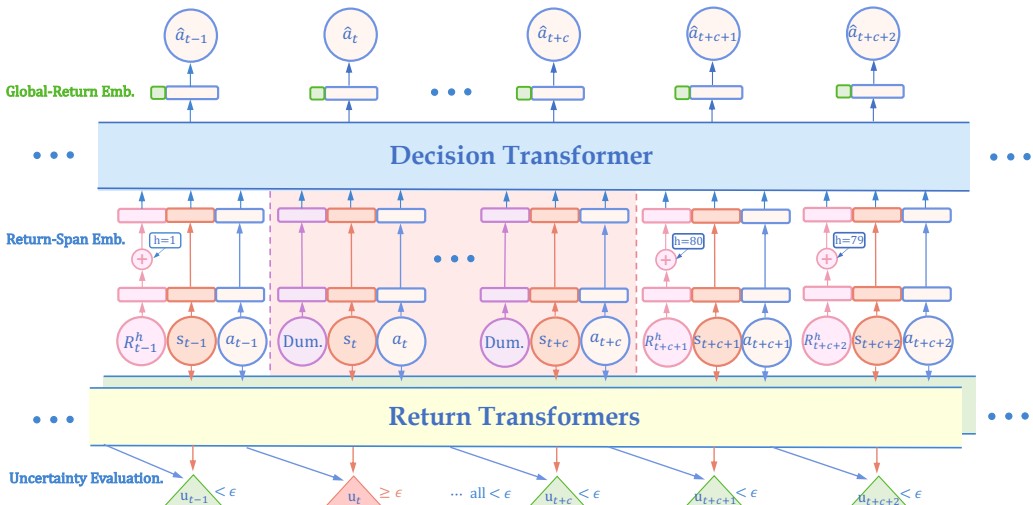

Figure 2: Overview of UNREST. **Lower:** Two return prediction transformers are trained for uncertainty estimation. The sequence is then segmented into certain (no background) and uncertain (orange background) parts w.r.t. estimated uncertainties, with 'certain parts' conditioned on returns to the next segmentation positions, and dummy tokens in 'uncertain parts'. **Upper:** The same architecture as DTs is used for action prediction, except that we add a return-span embedding to the truncated return embedding, and concatenate the discretized global return embedding to the transformer output.

Afterward, two models feed their respective outputs into variance networks (Kendall & Gal, 2017b) to predict Gaussian distributions $\mathcal{N}(\cdot)$ of returns, which are optimized by maximizing log-likelihoods:

$$p_{\varphi_a}(R_t|\tau_{<t}^{\text{ret}}) = \mathcal{N}\big(\mu_{\varphi_a}(\tilde{x}_{a_{t-1}}), \sigma_{\varphi_a}(\tilde{x}_{a_{t-1}})\big), \mathcal{L}_{\text{return}}(\varphi_a) = \mathbb{E}_{\tau \sim \mathcal{D}}\big[-\sum_{t=1}^{T} \log p_{\varphi_a}(R_t|\tau_{<t}^{\text{ret}})\big],$$
$$p_{\varphi_s}(R_t|\tau_{<t}^{\text{ret}}, s_t) = \mathcal{N}\big(\mu_{\varphi_s}(\tilde{x}_{s_t}), \sigma_{\varphi_s}(\tilde{x}_{s_t})\big), \mathcal{L}_{\text{return}}(\varphi_s) = \mathbb{E}_{\tau \sim \mathcal{D}}\big[-\sum_{t=1}^{T} \log p_{\varphi_s}(R_t|\tau_{<t}^{\text{ret}}, s_t)\big]. \tag{4}$$

Practically, the networks are implemented as ensembles, which together form Gaussian Mixture Models (GMM) (Mai et al., 2022) to better capture the return distribution as in App. A.2. Compared to challenging high-dimensional transition function learning and complex generative training, the return distributions can be effectively learned with fewer resources. Finally, the impact of the stochastic environmental transition (i.e. the conditional mutual information) is calculated through the Kullback-Leibler (KL) divergence (Joyce, 2011) between these two distributions, as a means to measure the environmental uncertainty at timestep $t$:

$$u_t = D_{\text{KL}}\big(p_{\varphi_a}(R_t|\tau_{<t}^{\text{ret}}), p_{\varphi_s}(R_t|\tau_{<t}^{\text{ret}}, s_t)\big), \tag{5}$$

where a larger divergence implies more influence on the return from the transition $(s_{t-1}, a_{t-1} \to s_t)$.

## 4.3 DECISION TRANSFORMER TRAINED ON SEGMENTED SEQUENCES

In this section, we step to aid DT training in stochastic driving environments. As discussed in Sec. 4.1, we expect UNREST to segment sequences into certain and uncertain parts according to uncertainty estimations and learn to perform aggressively or cautiously within them, respectively. To achieve this, we propose to replace the conditioning global returns with truncated returns in 'certain parts', which are less affected by the environment due to uncertainty accumulation (Prop. 1), thus reliably helping the planner generalize to higher returns after training. Otherwise, in 'uncertain parts', the seemingly high return actions may easily cause safety issues due to environmental uncertainty, thus we simply ignore the stochastic returns, setting conditions as dummy tokens for behavior cloning.

**Segmentation strategy** is therefore a crucial point to reinvent DTs. To distinguish between different levels of environmental stochasticity, we define an uncertainty threshold $\epsilon$. Next, we estimate uncertainties $u_t$ by Eq. 5 for each timestep and record those larger than $\epsilon$ as uncertain. The sequence is then divided into certain and uncertain parts according to these marked uncertain timesteps.

An intuitive example of our segmentation strategy is illustrated in the lower part of Fig. 2. Specifically, the 'certain part' begins at a timestep with uncertainty below the threshold $\epsilon$ and persists until segmentation occurs at an uncertain timestep. Subsequently, the 'uncertain part' commences with the

newly encountered uncertain timestep and encompasses subsequent ones until the final $c-1$ timesteps are all identified as certain. Since uncertain timesteps may occur intensively and intermittently over a particular duration of driving (e.g. at an intersection), the hyperparameter $c$ is introduced to avoid frequent switching between certain and uncertain parts, and ensure a minimum length of segmented sequences. Finally, we represent the segmented sequence as:

$$\tau^{\text{seg}} = (h_1, R_1^h, s_1, a_1, h_2, R_2^h, s_2, a_2, ..., h_T, R_T^h, s_T, a_T), \tag{6}$$

where the conditioning return is modified as $R_t^h = \sum_{k=0}^{h_t-1} r_{t+k}$, which only involves rewards in the next $h_t$ steps (called the return-span). In 'uncertain parts', $h$ is set as empty for meaningless conditions (i.e. the dummy tokens). In this way UNREST will learn to ignore the insignificant conditioned targets and directly imitate expert actions, instead of being misguided by the uncertain return information. In 'certain parts', the return-span $h$ is set to the number of timesteps to the next segmentation step, so that the conditioning return $R^h$ incorporates the maximum duration that does not include any uncertain timesteps. With this segmentation and condition design, we derive the following proposition:

**Proposition 1 (UNREST Alignment Bound)** *Assuming that the rewards obtained are determined by transitions $(s, a \rightarrow s')$ at each timestep and UNREST is perfectly trained to fit the expert demonstrations, then the discrepancy between target truncated returns and URNEST's rollout returns is bounded by a factor of environmental stochasticity and data coverage.*

It reveals that under natural assumptions in driving environments, UNREST can generalize to achieve high returns with bounded error at less stochastic states if expert demonstrations cover corresponding actions. The formal statement and proof of the proposition are left to App. B.

Notably, due to sequence segmentation, the truncated returns no longer necessarily encompass all time steps from the present to the end of the sequence, which necessitates the return-span as a condition to provide information about the count of timesteps involved in returns. This enables UNREST to learn to get return $R_t^h$ over future $h_t$ timesteps. Otherwise, the model may be confused by the substantial differences in the magnitude of return conditions with varying timestep lengths (Hu et al., 2023).

**Policy formulation:** Unlike return transformers, DT takes the segmented sequence $\tau^{\text{seg}}$ as input:

$$\tilde{x}_{s_t} \sim \mathcal{T}_\theta(..., x_{R_t^h}, x_{s_t}, x_{a_t}), \tag{7}$$
$$\text{where } x_{R_t^h} = f_\theta^{R^h}(R_t^h) + f_\theta^h(h_t), \ x_{s_t} = f_\theta^s(s_t), \ x_{a_t} = f_\theta^a(a_t).$$

Here we use similar notations as return transformers in Sec. 4.2 except that models are parameterized by $\theta$. In the above formulation, a *return-span embedding* $f_\theta^h(h_t)$ is added to the return embedding, which bears the semantic meaning of how many timesteps are involved in the target return. Besides, to get the action distribution, the non-truncated *global-return embedding* $f_\theta^R(R_t)$ is optionally added to the output $\tilde{x}_{s_t}$ to provide additional longer horizon guidance for planning. Using $[\cdot \| \cdot]$ to denote the concatenation of two vectors along the last dimension, the final predicted action distribution is:

$$\pi_\theta(a_t|\tau_{<t}^{\text{seg}}, h_t, R_t, R_t^h, s_t) = \mathcal{N}\big(\mu_\theta([\tilde{x}_{s_t} \| f_\theta^R(R_t)]), \sigma_\theta([\tilde{x}_{s_t} \| f_\theta^R(R_t)])\big). \tag{8}$$

The learning objective is modified from Eq. 2 of DT, with detailed training process in App. C:

$$\mathcal{L}_{\text{UNREST}}(\theta) = \mathbb{E}_{\tau^{\text{seg}} \sim \mathcal{D}}\big[ -\sum_{t=1}^T \log \pi_\theta(a_t|\tau_{<t}^{\text{seg}}, h_t, R_t, R_t^h, s_t)\big]. \tag{9}$$

### 4.4 Uncertainty-guided Planning

During inference, we still differentiate between certain and uncertain states to account for the impacts of environmental stochasticity. To enable real-time prediction of environmental stochasticity, we introduce a lightweight uncertainty prediction model $u_\psi(\cdot)$. At each timestep, we query the predictor to obtain the uncertainty measure. If the current state transition is highly uncertain, we set the conditioning target to a dummy token to facilitate cautious planning, consistent with training. Conversely, at states with certain transitions, the planner acts aggressively to attain high target returns. The predictor can be implemented by various approaches, like a neural network or a heuristic estimator, whose results are summarized in Tab. 8 and 9. Practically, we choose the KD-Tree (Redmond & Heneghan, 2007) for its high computational efficiency and favorable estimation performance, with states as tree nodes and uncertainties estimated by return transformers as node values.

Different from the conventional planning procedure of DTs, UNREST requires the specification of not only the target global return $R_1$, but also the initial return-span $h_1 = H$ and the target truncated

return $R_1^h$. After segmentation, the effective planning horizon of the trained sequences is reduced to the return-span $h_t$. Once $h_t$ reaches 1, we need to reset the target return and the return-span. Practically, we simply reset $h_t$ to a fixed return horizon $H$. For the truncated return, we train a return prediction model $R_\varphi^h(\cdot)$ similar to that defined in Eq. 4 and take the upper percentile $\eta$ of the predicted distribution as the new target return to attain. The hyperparameter $\eta$ can be tuned for a higher target return. We do not need to consider targets at 'uncertain states' since they are just set as dummy tokens. The complete planning process is summarized in Alg. 1.

## 5 EXPERIMENTS

In this section, we conduct extensive experiments to answer the following questions. **Q1:** How does UNREST perform in different driving scenarios? **Q2:** How do components of UNREST influence its overall performance? **Q3:** Does our uncertainty estimation possess interpretability?

### 5.1 EXPERIMENT SETUP

In this section, we briefly describe the setup components of our experiments. Please find more details of model implementation, training, and evaluation processes in App. E.

**Datasets:** The offline dataset $\mathcal{D}$ is collected from the CARLA simulator (Dosovitskiy et al., 2017) with its built-in Autopilot. Specifically, we collect 30 hours of data from 4 towns (Town01, Town03, Town04, Town06) under 4 weather conditions (ClearNoon, WetNoon, HardRainNoon, ClearSunset), saving tuple $(s_t, a_t, r_t)$ at each timestep. More details about the state, action compositions, and our reward definitions are left to App. D.

---

**Algorithm 1:** Uncertainty-guided planning

**Input:** History $\tau$, return horizon $H$, state $s_t$, policy $\pi_\theta$, uncertainty model $u_\psi$, return model $R_\varphi^h$, return percentile $\eta$.

1   # First, update target returns.
2   Update target global return $R_t \leftarrow R_{t-1} - r_{t-1}$;
3   **if** $h_{t-1} == 1$ or $u_\psi(s_{t-1})$ is True **then**
4      Reset return span to return horizon $h_t \leftarrow H$;
5      Predict target return $R_t^h \leftarrow R_\varphi^h(\tau, h_t, s_t, \eta)$;
6   **else**
7      Update return span $h_t \leftarrow h_{t-1} - 1$;
8      Update target return $R_t^h \leftarrow R_{t-1}^h - r_{t-1}$;
9   # Second, evaluate new state is uncertain or not.
10   **if** $u_\psi(s_t)$ is True **then**
11      Reset conditioning target $(h_t, R_t^h) \leftarrow \emptyset$;
12   # Finally, select action using trained policies.
13   Sample and take action
     $a_t \sim \pi_\theta(a_t | \tau, h_t, R_t, R_t^h, s_t)$;
14   Update history $\tau \leftarrow \tau \cup (h_t, R_t, R_t^h, s_t, a_t)$.
**Output:** Action $a_t$ and history $\tau$.

---

**Metrics:** We evaluate models at training and new driving scenarios and report metrics from the CARLA challenge (Dosovitskiy et al., 2017; team, 2020) to measure planners' driving performance: infraction score, route completion, success rate, and driving score. Besides, as done in (Hu et al., 2022), we also report the normalized reward (the ratio of total return to the number of timesteps) to reflect driving performance at timestep level. Among them, the driving score is the most significant metric that accounts for various indicators like driving efficiency, safety, and comfort.

**Baselines:** First, we choose two IL baselines: vanilla Behavior Cloning (BC) and Monotonic Advantage Re-Weighted Imitation Learning (MARWIL) (Wang et al., 2018). Apart from IL baselines, we also include state-of-the-art offline RL baselines: Conservative Q-Learning (CQL) (Kumar et al., 2020), Implict Q-Learning (IQL) (Kostrikov et al., 2021). Constraints-Penalized Q-Learning (CPQ) (Xu et al., 2022) is chosen as a safe (cautious) offline RL baseline. Besides, we select two classic Transformer-based offline RL algorithms: Decision Transformer (DT) (Chen et al., 2021) and Trajectory Transformer (TT) (Janner et al., 2021) as baselines. Finally, we adopt three algorithms as rigorous baselines: Separated Latent Trajectory Transformer (SPLT) (Villaflor et al., 2022), Environment-Stochasticity-Independent Representations (ESPER) (Paster et al., 2022), and Dichotomy of Control (DoC) (Yang et al., 2022). These algorithms fit state transition models and employ generative training to mitigate DTs' limitations in stochastic environments.

### 5.2 DRIVING PERFORMANCE

Firstly, we implement all the models, then evaluate their performance at training (Town03) and new (Town05) scenarios (**Q1**), whose results are summarized in Tab. 1.

Analyzing the results, we first notice that DT (Chen et al., 2021) performs worst among all sequence models, with a significant gap compared to the simple offline RL baseline IQL (Kostrikov et al., 2021).

Table 1: Driving performance on train (new) town and train (new) weather conditions in CARLA. Mean and standard deviation are computed over 3 seeds. All metrics are recorded in percentages (%) except the normalized reward. The best results are in bold and our method is colored in gray.

| Planner | Driving Score↑ | Success Rate↑ | Route Completion↑ | Infraction Score↑ | Normalized Reward↑ |
|---|---|---|---|---|---|
| BC | 51.9 ± 1.9 (45.6 ± 5.2) | 37.9 ± 4.7 (36.3 ± 8.6) | 79.7 ± 6.0 (77.1 ± 7.8) | 54.5 ± 1.8 (47.0 ± 4.8) | 0.63 ± 0.02 (0.61 ± 0.04) |
| MARWIL (Wang et al., 2018) | 54.3 ± 2.0 (47.8 ± 3.4) | 44.8 ± 3.2 (42.2 ± 2.2) | 81.4 ± 3.2 (80.2 ± 3.8) | 57.9 ± 2.0 (48.4 ± 4.4) | 0.65 ± 0.02 (0.63 ± 0.01) |
| CQL (Kumar et al., 2020) | 55.0 ± 2.4 (50.7 ± 2.8) | 48.6 ± 4.5 (42.8 ± 4.6) | 80.5 ± 3.4 (78.7 ± 5.2) | 62.4 ± 3.0 (57.7 ± 5.0) | 0.65 ± 0.03 (0.62 ± 0.02) |
| IQL (Kostrikov et al., 2021) | 55.9 ± 3.3 (52.2 ± 3.3) | 50.2 ± 6.2 (44.2 ± 2.2) | 77.2 ± 4.6 (68.8 ± 4.2) | 68.4 ± 2.4 (62.6 ± 6.2) | 0.66 ± 0.03 (0.60 ± 0.01) |
| CPQ (Xu et al., 2022) | 54.7 ± 3.0 (53.8 ± 2.7) | 46.6 ± 3.5 (40.3 ± 5.7) | 79.4 ± 4.2 (78.0 ± 4.6) | 66.9 ± 2.8 (64.4 ± 4.2) | 0.63 ± 0.02 (0.64 ± 0.02) |
| DT (Chen et al., 2021) | 55.2 ± 2.0 (47.6 ± 1.2) | 48.0 ± 4.7 (46.4 ± 0.3) | 82.6 ± 1.0 (81.8 ± 4.9) | 57.4 ± 1.3 (47.3 ± 3.4) | 0.66 ± 0.01 (0.64 ± 0.02) |
| TT (Janner et al., 2021) | 58.3 ± 3.3 (54.8 ± 2.2) | 45.9 ± 5.2 (52.0 ± 4.4) | 78.4 ± 5.8 (77.0 ± 4.6) | 63.6 ± 3.6 (56.0 ± 5.0) | **0.74 ± 0.02** (0.64 ± 0.03) |
| SPLT (Villaflor et al., 2022) | 57.8 ± 4.9 (56.4 ± 6.1) | 23.1 ± 8.1 (39.5 ± 9.5) | 36.7 ± 8.7 (48.2 ± 9.7) | **73.9 ± 1.4** (**70.7 ± 8.3**) | 0.55 ± 0.03 (0.57 ± 0.05) |
| ESPER (Paster et al., 2022) | 54.8 ± 2.0 (51.2 ± 3.1) | 48.5 ± 4.3 (53.3 ± 2.0) | 77.3 ± 5.4 (84.7 ± 1.4) | 56.2 ± 2.2 (46.9 ± 5.6) | 0.64 ± 0.04 (0.63 ± 0.02) |
| DoC (Yang et al., 2022) | 56.9 ± 3.1 (54.4 ± 2.3) | 49.9 ± 5.2 (54.0 ± 4.2) | 79.2 ± 3.6 (84.0 ± 2.8) | 60.3 ± 2.4 (51.5 ± 4.8) | 0.64 ± 0.03 (0.65 ± 0.04) |
| UNREST | **63.5 ± 3.2** (**62.9 ± 4.0**) | **54.5 ± 7.0** (**57.5 ± 5.4**) | **83.8 ± 3.1** (**90.0 ± 6.0**) | 70.2 ± 2.8 (62.9 ± 3.8) | 0.64 ± 0.04 (**0.65 ± 0.03**) |
| Expert (Dosovitskiy et al., 2017) | 74.0 ± 6.0 (75.3 ± 1.3) | 65.4 ± 8.8 (68.6 ± 5.1) | 84.2 ± 4.6 (95.8 ± 1.1) | 82.8 ± 3.2 (77.5 ± 1.6) | 0.72 ± 0.01 (0.69 ± 0.01) |

(a) Changing lane: Failing case of TT

(b) Changing lane: UNREST's performance

(c) Passing green light: Failing case of SPLT

(d) Passing green light: UNREST's performance

Figure 3: UNREST performs well at failing cases of TT and SPLT. White rectangles are ego-vehicles.

In new scenarios, it even shows a similar performance to BC. We attribute this to the uncertainty of global return it conditions on. When the conditioning target does not reflect the true outcomes of agent actions, DT learns to ignore the target return condition and makes decisions solely based on the current state like BC. Furthermore, ESPER (Paster et al., 2022) and DoC (Yang et al., 2022) also perform poorly in both scenarios, which may be a result of ineffective adversarial and VAE training of long-horizon and complex driving demonstrations.

To tackle environmental stochasticity, SPLT learns to predict the worst state transitions and achieves the best infraction score. However, its overly cautious planning process leads it to stand still in many scenarios like Fig. 3(c), resulting in an extremely poor route completion rate and normalized reward. TT instead learns a transition model regardless of environmental stochasticity and behaves aggressively. It attains the highest normalized reward at training scenarios since the metric prioritizes planners that rapidly move forward (but with low cumulative reward because of its short trajectory length caused by frequent infractions). In unseen driving scenarios, TT often misjudges the speed of preceding vehicles, resulting in collisions and lower normalized rewards (than ours) like in Fig. 3(a).

Notably, UNREST achieves the highest driving score, route completion rate, success rate in both seen and unseen scenarios, and highest normalized reward in new scenarios without the need to learn transition or complex generative models. For the driving score, it surpasses the strongest baselines, TT (Janner et al., 2021) over 5% in training scenarios, and SPLT (Villaflor et al., 2022) over 6% in new scenarios (in absolute value), achieving a reasonable balance between aggressive and cautious behavior. For all the metrics, the results demonstrate that UNREST obtains significant improvement in terms of safety, comfort, and efficiency, and effectively increases the success rate of driving tasks. It demonstrates that the truncated returns successfully mitigate impacts of stochasticity and provide effective action supervision. In App. F.2, we test and find that UNREST only occupies slightly more resources than DT, and runs significantly faster while consuming less space than TT and SPLT, which additionally fit state transition models.

## 5.3 ABLATION STUDY

The key components of UNREST include global return embedding, return-span embedding, ensemble-based uncertainty estimation, and the uncertainty-guided planning process. In this section, we conduct ablation experiments by separately removing these components to explore their impacts on the overall performance of UNREST (Q2). Results are shown in Tab. 2 and Tab. 3.

The ablation results show pronounced differences, and it is apparent that the elimination of any part of the three components leads to a decline in UNREST's driving score in new scenarios. Among

Table 2: Ablation study results for UNREST on train town and train weather conditions.

| Planner | Driving Score↑ | Success Rate↑ | Route Co.↑ | Infraction Score↑ | Norm. Reward↑ |
|---|---|---|---|---|---|
| W/o global emb. | **64.5 ± 2.8** | **55.8 ± 6.0** | 80.2 ± 4.2 | 68.0 ± 1.6 | 0.63 ± 0.03 |
| W/o ret-span emb. | 57.0 ± 1.8 | 33.3 ± 4.5 | 47.6 ± 3.3 | 65.6 ± 3.1 | 0.57 ± 0.01 |
| W/o ensemble | 60.4 ± 3.4 | 51.3 ± 6.3 | 80.2 ± 4.4 | 65.5 ± 3.1 | 0.62 ± 0.03 |
| W/o guided plan | 55.1 ± 2.3 | 42.1 ± 1.5 | 52.6 ± 6.0 | 65.2 ± 1.7 | 0.57 ± 0.02 |
| Full model | 63.5 ± 3.2 | 54.5 ± 7.0 | **83.8 ± 3.1** | **70.2 ± 2.8** | **0.64 ± 0.04** |

Table 3: Ablation study results for UNREST on new town and new weather conditions.

| Planner | Driving Score↑ | Success Rate↑ | Route Co.↑ | Infraction Score↑ | Norm. Reward↑ |
|---|---|---|---|---|---|
| W/o global emb. | 61.8 ± 3.3 | 55.0 ± 3.8 | 82.8 ± 3.4 | 57.8 ± 2.7 | 0.64 ± 0.03 |
| W/o ret-span emb. | 56.2 ± 4.7 | 47.6 ± 3.7 | 78.9 ± 5.4 | 58.2 ± 3.4 | 0.59 ± 0.02 |
| W/o ensemble | 57.4 ± 2.8 | 48.8 ± 4.1 | 85.4 ± 3.8 | 56.7 ± 2.7 | 0.61 ± 0.04 |
| W/o guided plan | 54.0 ± 3.0 | 54.5 ± 2.7 | 81.8 ± 3.8 | 57.3 ± 3.1 | 0.59 ± 0.03 |
| Full model | **62.9 ± 4.0** | **57.5 ± 5.4** | **90.0 ± 6.0** | **62.9 ± 3.8** | **0.65 ± 0.03** |

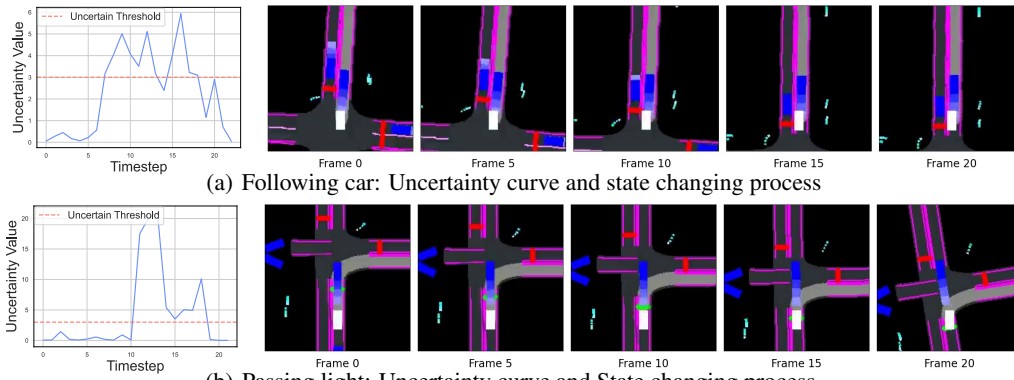

(a) Following car: Uncertainty curve and state changing process

(b) Passing light: Uncertainty curve and State changing process

Figure 4: Visualizations of UNREST's uncertainty estimation results.

them, the global return embedding has the slightest impact, which suggests that the highly uncertain global return may not provide effective guidance, or that the truncated return is already sufficient for making reasonable decisions (temporal locality, Prop. 2). When the return-span embedding is removed, the absolute driving score drops by about 6%. This implies that the introduction of return-span embedding provides necessary information about the timesteps needed to achieve the target return. Removing the ensemble of return transformers (i.e. the GMM) induces a significant performance drop in both scenarios. It shows that a simple Gaussian distribution cannot well express the return distribution, resulting in poor uncertainty calibration capability (corresponding with results in Fig. 6). Finally, after canceling uncertainty estimation at test time, the driving and infraction scores of UNREST drop dramatically, which proves the importance of cautious planning.

## 5.4 UNCERTAINTY VISUALIZATION

Finally, We verify the interpretability of UNREST's uncertainty estimation through visualizations (Q3). Typically, in Fig. 4(a) we observe an initial increase in uncertainty as the ego-vehicle enters the lane to follow another vehicle, owing to the lack of knowledge about the other vehicle's behavior. This uncertainty gradually decreases back below the threshold when the vehicle stabilizes in the following state. Fig. 4(b) shows a green light crossing scenario. While approaching the green light, the uncertainty about the light state causes the uncertainty to rise quickly. After the vehicle has moved away from the traffic light, the uncertainty immediately drops back below the threshold.

## 6 CONCLUSION: SUMMARY AND LIMITATIONS

The paper presents UNREST, an uncertainty-aware decision transformer to apply offline RL in stochastic driving environments. Specifically, we propose a novel uncertainty measurement by computing the divergence of return prediction models, bypassing complex transition and generative training. Then based on properties we discover at driving, we segment sequences w.r.t. estimated uncertainty and adopt truncated returns as conditioning goals. This new condition helps UNREST learn policies that are less affected by stochasticity. Dynamic uncertainty estimation is also integrated at inference for cautious planning. Empirical results demonstrate UNREST's superior performance, lower resource occupation, and effective uncertainty estimation in various driving scenarios.

One limitation of this work is that the inference process of UNREST is somewhat complex with auxiliary return, uncertainty estimation models, and hyperparameters. One possible direction for improvement is to integrate return and uncertainty predictions into the model architecture, which we leave for future work. Although this work is evaluated in the CARLA simulator, we believe the proposed framework can surmount the sim-to-real gap and benefit practical autonomous driving.

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

## A    ADDITIONAL RELATED WORKS

### A.1    PLANNING FOR SELF-DRIVING

Most autonomous driving systems (Montemerlo et al., 2008; Urmson et al., 2008; Ziegler et al., 2014) adopt a modular pipeline to break down the massive driving task into a set of submodules, with planning being one of the most fundamental components. The objective of motion planning is to efficiently drive the ego-vehicle to the destination while conforming to safety and comfort constraints, which is essentially a decision problem. Often engineers manually design rules (Ziegler et al., 2014; Chen et al., 2019b) for specific driving scenarios, which struggles to scale for complex tasks. In contrast, IL is the earliest (Pomerleau, 1988) and most widely (Bansal et al., 2018; Zeng et al., 2019; Hu et al., 2022; Bronstein et al., 2022; Codevilla et al., 2019; Zhang et al., 2021) used learning-based planning algorithm, which learns from offline data for either a cost function (Zeng et al., 2019) or an executable policy (Bansal et al., 2018; Hu et al., 2022; Bronstein et al., 2022; Codevilla et al., 2019). However, they are strictly limited by expert quality (Ross et al., 2011) and will perform poorly when encountering out-of-distribution (OOD) states (Zhang et al., 2021; Ross et al., 2011; Prakash et al., 2020). RL algorithms (Chen et al., 2019a; 2020) own better performance and generalizability with the downside of trial-and-error learning. To address these shortcomings, we employ offline RL as an alternative to train our planner.

### A.2    UNCERTAINTY ESTIMATION

Variance networks estimate uncertainty by loss attenuation (Kendall & Gal, 2017b). Treating the output as a Gaussian distribution $\mathcal{N}(\cdot)$, given input $x$ and its label $y$, the network outputs mean $\mu(x)$ and variance $\sigma^2(x)$ at its final layer. The network parameter $\phi$ is subsequently optimized by maximizing sample log-likelihood:

$$\mathcal{L}_{\text{varnet}}(\phi) = \mathbb{E}_{(x,y)\sim\mathcal{D}}\big[-\log\mathcal{N}_\phi(y|x)\big] = \mathbb{E}_{(x,y)\sim\mathcal{D}}\left[\frac{\big((\mu_\phi(x)-y^2\big)}{\sigma_\phi^2(x)} + \ln\sigma_\phi^2(x)\right]. \quad (10)$$

As the variance predicted by neural networks may be not well-calibrated or overestimated (Kuleshov et al., 2018), in this paper we practically train an ensemble of $K$ variance networks with data drawn from different subsets of the dataset (implemented by masking) and estimate the variance according to (Mai et al., 2022):

$$\sigma^2 = \sum_{l=1}^K \sigma_l^2 + \mu_l^2 - \mu^2, \text{ where } \mu = \sum_{l=1}^K \mu_k/K, \quad (11)$$

which has been proven to have strong estimation capability (Mai et al., 2022).

## B    PROOF FOR PROPOSITION 1

We next formally state our theorem, followed by detailed proofs of the proposition.

### B.1    THEOREM STATEMENT

Before stepping into the main theorem, we first introduce the problem setting that we undertake for the sake of concise proof. First, we assume the reward $r_t$ is within $[0, 1]$ at each timestep for conveniently bounding the error. Besides, we use $g(s_1, a_{1:h})$ to denote the cumulative rewards by rolling out the open loop sequence of actions $a_{1:h}$ under the deterministic dynamics $P : \mathcal{S} \times \mathcal{A} \to \mathcal{S}$ and reward function $r : \mathcal{S} \times \mathcal{A} \to [0, 1]$ (so that the maximum return difference over future $h$ timesteps is $h$). Moreover, $J_h(\pi)$ represents the rollout rewards of policy $\pi$ over future $h$ timesteps. Next, we formally introduce the assumption, theorem and its corresponding proof, which is inspired by the reference (Brandfonbrener et al., 2022):

**Theorem 1 (UNREST Alignment Bound)** *Consider an MDP* $(\mathcal{S}, \mathcal{A}, P, r, \gamma)$*, expert behavior* $\beta : \mathcal{S} \times \mathcal{A} \to [0, 1]$ *and conditioning function* $f : \mathcal{S} \times \mathbb{N}^+ \to \mathbb{R}$*. Assume the following:*

1. *Return Coverage:* $p_\beta(g = f(s_1, h)|s_1) \geq \alpha_f$ *for the initial state* $s_1$ *and return-span h.*

2. *Near Determinism:* $p(r \neq r(s, a) \text{ or } s' \neq P(s, a)|s, a) \leq \delta$ *at all state-action pairs* $(s, a)$ *for dynamics P and reward function r.*

3. *Consistency of $f$: $f(s,h)=f(s',h-1)$ for all states $s$.*

*Then (URT shorted for UNREST):*

$$\mathbb{E}_{s_1,h}\big[f(s_1,h)\big] - J_h(\pi_f^{URT}) \le \delta(\frac{1}{\alpha_f} + 2)h^2. \tag{12}$$

As shown by the theorem, the error between the specified target and UNREST's rollout return is bounded by the factor of environment determinism $\delta$, data coverage $\alpha$, and the horizon $h$.

Based on the theorem, we aim to demonstrate the generalization ability of UNREST at the identified 'certain states' by our uncertainty estimation stra. Specifically, we have the following lemma:

**Lemma 1 (Determinism Equality)** *Assuming that the rewards obtained are uniquely determined by transitions $(s, a \to s')$ at each timestep, then there exists $\epsilon > 0$ and $\delta > 0$, such that:*

$$D_{KL}\big[p(R_t|\tau_{<t}) \,||\, p(R_t|s_t, \tau_{<t})\big] \le \epsilon \iff p(r \ne r(s,a) \text{ or } s' \ne P(s,a)|s,a) \le \delta. \tag{13}$$

While the reverse direction is easy to see (the state transition probability is nearly deterministic so the transitions cannot provide additional information for return prediction), here we focus on proving the forward direction. Let us demonstrate its contrapositive proposition:

If $p(r \ne r(s,a) \text{ or } s' \ne P(s,a)|s,a) > \delta$, then $\exists\epsilon$, s.t. $D_{KL}\big[p(R_t|\tau_{<t}) \,||\, p(R_t|s_t, \tau_{<t})\big] > \epsilon$.
$$\tag{14}$$

First, since $r$ is uniquely (different transitions lead to distinct rewards) determined by $(s, a \to s')$, we can induce that $r \ne r(s,a)$ is equivalent with $s' \ne P(s,a)$ as the only difference can occur in $s'$. Therefore, we need to prove:

If $p(r \ne r(s,a)|s,a) > \delta$, then $\exists\epsilon$, s.t. $D_{KL}\big[p(R_t|\tau_{<t}) \,||\, p(R_t|s_t, \tau_{<t})\big] > \epsilon$. $\tag{15}$

Considering the extreme case, set $\gamma \to 0$, thus we have $R_t \to r_t$. From the perspective of proof by contradiction, the KL-divergence cannot be smaller than any $\epsilon > 0$, since $p(r_t|\tau_{>t})$ and $p(r_t|s_t, \tau_{>t})$ must differ for that the probability $r_t \ne r(s_{t-1}, a_{t-1})$ is larger than 0 and can be determined by $s_t$. To this end, we have proved the contrapositive proposition of the forward direction, which is equivalent to the original proposition.

Therefore, we can summarize from the above theorem and lemma that UNREST can generalize to achieve high returns at 'certain states' as long as the corresponding actions are covered by the expert.

### B.2 PROOF FOR THE THEOREM

The next theorem proof is largely built on (Brandfonbrener et al., 2022). Note that we omit the superscript 'URT' to simplify the equations. First, we expand the left term in Eq. 12:

$$
\begin{aligned}
\mathbb{E}_{s_1,h}\big[f(s_1,h)\big] - J_h(\pi_f) &= \mathbb{E}_{s_1}\Big[\mathbb{E}_{\pi_f|s_1}\big[f(s_1,h) - g_1\big]\Big] \\
&= \mathbb{E}_{s_1,h}\Big[\mathbb{E}_{a_{1:h}\sim\pi_f|s_1}\big[f(s_1,h) - g(s_1, a_{1:h})\big]\Big] \\
&\quad + \mathbb{E}_{s_1,h}\Big[\mathbb{E}_{a_{1:h}\sim\pi_f|s_1}\big[g(s_1, a_{1:h}) - g_1\big]\Big] \\
&\le \mathbb{E}_{s_1,h}\Big[\mathbb{E}_{a_{1:h}\sim\pi_f|s_1}\big[f(s_1,h) - g(s_1, a_{1:h})\big]\Big] + \delta h^2.
\end{aligned}
\tag{16}
$$

The last step follows by bounding the difference between $g_1$ and $g(s_1, a_{1:h})$ by the maximum return difference $h$ and a union of probability bound over $h$ timesteps based on the near determinism assumption:

$$h \cdot \sup_{s_1} \bigcup_{t}^{h} p_{a_t\sim\pi_f|s_1}\big(r_t \ne r(s_t, a_t) \text{ or } s_{t+1} \ne P(s_t, a_t)\big) \le \delta h^2. \tag{17}$$

For the first term in Eq. 16, it can also be expressed and bounded by the maximum return difference:

$$\mathbb{E}_{s_1,h}\Big[\mathbb{E}_{a_{1:h}\sim\pi_f|s_1}\big[f(s_1,h) - g(s_1, a_{1:h})\big]\Big] \le \mathbb{E}_{s_1,h}\int_{a_{1:h}} p_{\pi_f}(a_{1:h}|s_1)\mathbb{I}\big[g(s_1, a_{1:h}) \ne f(s_1,h)\big]h.$$
$$\tag{18}$$

To bound this term, we need to further expand the distribution $p_{\pi_f}$. First, with the assumption that UNREST is perfectly fitted to the expert dataset, we can deduce the following by the bayesian law:

$$\pi_f(a_1|s_1) = \beta(a_1|s_1)\frac{p_\beta(g_1 = f(s_1, h)|s_1, a_1)}{p_\beta(g_1 = f(s_1, h)|s_1)}. \tag{19}$$

For simplification, we use $\bar{s}_t = P(s_1, a_{1:t-1})$ to denote the state reached by following the deterministic dynamics defined by $P$ till timestep $t$. Next, based on the near determinism, consistency and coverage assumption, we can expand $p_{\pi_f}$ to get:

$$p_{\pi_f}(a_{1:h} \mid s_1) = \pi_f(a_1 \mid s_1) \int_{s_2} p(s_2 \mid s_1, a_1) \, p_{\pi_f}(a_{2:h} \mid s_1, s_2)$$

$$\leq \pi_f(a_1 \mid s_1) \, p_{\pi_f}(a_{2:h} \mid s_1, \bar{s}_2) + \delta$$

$$= \beta(a_1 \mid s_1) \frac{p_\beta(g_1 = f(s_1, h) \mid s_1, a_1)}{p_\beta(g_1 = f(s_1, h) \mid s_1)} p_{\pi_f}(a_{2:h} \mid s_1, \bar{s}_2) + \delta$$

$$\leq \beta(a_1 \mid s_1) \frac{\delta + p_\beta(g_1 - r(s_1, a_1) = f(s_1, h) - r(s_1, a_1) \mid s_1, a_1, \bar{s}_2)}{p_\beta(g_1 = f(s_1, h) \mid s_1)} p_{\pi_f}(a_{2:h} \mid s_1, \bar{s}_2) + \delta$$

$$= \beta(a_1 \mid s_1) \frac{\delta + p_\beta(g_2 = f(\bar{s}_2, h-1) \mid \bar{s}_2)}{p_\beta(g_1 = f(s_1, h) \mid s_1)} p_{\pi_f}(a_{2:h} \mid s_1, \bar{s}_2) + \delta$$

$$\leq \beta(a_1 \mid s_1) \frac{p_\beta(g_2 = f(\bar{s}_2, h-1) \mid \bar{s}_2)}{p_\beta(g_1 = f(s_1, h) \mid s_1)} p_{\pi_f}(a_{2:h} \mid s_1, \bar{s}_2) + \delta\left(\frac{1}{\alpha_f} + 1\right). \tag{20}$$

Similarly, we can further expand $p_{\pi_f}(a_{2:h} \mid s_1, \bar{s}_2)$ using the same rule:

$$p_{\pi_f}(a_{2:h} \mid s_1, \bar{s}_2) = \pi_f(a_2 \mid \bar{s}_2) \int_{s_3} p(s_3 \mid \bar{s}_2, a_2) p_{\pi_f}(a_{3:h} \mid s_1, \bar{s}_2, s_3)$$

$$\leq \pi_f(a_2 \mid \bar{s}_2) p_{\pi_f}(a_{3:h} \mid s_1, \bar{s}_2, \bar{s}_3) + \delta$$

$$= \beta(a_2 \mid \bar{s}_2) \frac{p_\beta(g_2 = f(\bar{s}_2, h-1)|\bar{s}_2, a_2)}{p_\beta(g_2 = f(\bar{s}_2, h-1)|\bar{s}_2)} p_{\pi_f}(a_{3:h} \mid s_1, \bar{s}_2, \bar{s}_3) + \delta. \tag{21}$$

Substituting this back to Eq. 20, we have:

$$p_{\pi_f}(a_{1:h} \mid s_1)$$

$$\leq \beta(a_1 \mid s_1)\beta(a_2 \mid \bar{s}_2) \frac{p_\beta(g_2 = f(\bar{s}_2, h-1) \mid \bar{s}_2)}{p_\beta(g_1 = f(s_1, h) \mid s_1)} \cdot \frac{p_\beta(g_2 = f(\bar{s}_2, h-1) \mid \bar{s}_2, a_2)}{p_\beta(g_2 = f(\bar{s}_2, h-1) \mid \bar{s}_2)} p_{\pi_f}(a_{3:h} \mid s_1, \bar{s}_2, \bar{s}_3)$$

$$+ 2\delta\left(\frac{1}{\alpha_f} + 1\right)$$

... recursively expand $p_{\pi_f}$

$$\leq \prod_{t=1}^{h} \beta(a_t \mid \bar{s}_t) \frac{p_\beta(g_h = f(\bar{s}_h, 1) \mid \bar{s}_h, a_h)}{p_\beta(g_1 = f(s_1, h) \mid s_1)} + h\delta\left(\frac{1}{\alpha_f} + 1\right)$$

$$= \prod_{t=1}^{h} \beta(a_t \mid \bar{s}_t) \frac{\mathbb{I}\left[g(s_1, a_{1:h}) = f(s_1, h)\right]}{p_\beta(g_1 = f(s_1, h) \mid s_1)} + h\delta\left(\frac{1}{\alpha_f} + 1\right). \tag{22}$$

The last step is deduced by the trajectory determinism and the consistency of the conditioning function $f$. Multiplying this back to Eq. 18 and noticing that the two indicator functions can never both be 1, we can yield that:

$$\mathbb{E}_{s_1, h}\left[\mathbb{E}_{a_{1:h} \sim \pi_f|s_1}\left[f(s_1, h) - g(s_1, a_{1:h})\right]\right] \leq h^2 \delta(\frac{1}{\alpha_f} + 1). \tag{23}$$

This can finally yield the bound in Eq. 12 by adding back to Eq. 16.

## C UNREST TRAINING PROCEDURE

To get a more clear understanding of our training pipeline, we present the training procedure of UNREST in Alg. 2.

---

**Algorithm 2:** UNREST training procedure

---

**Input:** Offline dataset $\mathcal{D} = \{\tau_i\}_{i=1}^N$, batch size $B$.

1    # Stage I: Train return transformers.
2    **for** each iteration **do**
3       Sample batch $\mathcal{B} = \{\tau_i\}_{i=1}^B$ from $\mathcal{D}$, where $\tau_i = \{(s_t, a_t)\}_{t=1}^T$;
4       Feed sampled sequences into Transformer and get embeddings $\tilde{x}_{s_t}, \tilde{x}_{a_t}$ by Eq. 3;
5       Feed Transformer outputs into variance networks to predict return
       $R_{\varphi_a}(R_t|\tau_{<t}), R_{\varphi_s}(R_t|\tau_{<t}, s_t)$;
6       Update network parameters based on the learning objectives in Eq. 4;

7    # Stage II: Segment sequence w.r.t. estimated uncertainty.
8    Compute uncertainty of each timestep $u_t \leftarrow D_{\mathrm{KL}}(R_{\varphi_a}(R_t|\tau_{<t}), R_{\varphi_s}(R_t|\tau_{<t}, s_t))$;
9    Segment sequence w.r.t. uncertainties $\mathcal{D}^{\mathrm{seg}} \leftarrow \{\tau_i^{\mathrm{seg}}\}_{i=1}^N$, as discussed in Sec. 4.3;
10   # Stage III: Train decision model.
11   **for** each iteration **do**
12      Sample batch $\mathcal{B} = \{\tau_i^{\mathrm{seg}}\}_{i=1}^B$ from $\mathcal{D}^{\mathrm{seg}}$, where $\tau_i^{\mathrm{seg}} = \{(h_t, R_t, R_t^h, s_t, a_t)\}_{t=1}^T$;
13      Feed sampled sequences into Transformer and get embeddings $\tilde{x}_{R_t^h}, \tilde{x}_{s_t}, \tilde{x}_{a_t}$ by Eq. 7;
14      **if** using global return **then**
15        Concatenate Transformer outputs with discretized global return $\tilde{x}_{s_t} \leftarrow [\tilde{x}_{s_t} \,\|\, f^R(R_t)]$;
16      Feed $\tilde{x}_{s_t}$ into variance network to predict action distribution $\pi_\theta(a_t|\tau_{<t}^{\mathrm{seg}}, h_t, R_t, R_t^h, s_t)$;
17      Update network parameters based on learning objective in Eq.9;

**Output:** Trained return transformers $R_{\varphi_a}(\cdot), R_{\varphi_s}(\cdot)$ and policy $\pi_\theta(\cdot)$.

---

## D    DATASET INFORMATION

Our dataset is collected from the CARLA simulator, using its built-in Autopilot, which is a rule-based motion planner. Specifically, CARLA (Dosovitskiy et al., 2017) is an open-sourced simulator designed for autonomous driving research. It provides researchers with a realistic environment that faithfully simulates traffic dynamics, weather conditions, and high-fidelity sensor data. The pre-built scenarios in CARLA offer a diverse set of driving scenes, while its high level of customizability and flexibility make it the preferred simulation platform for a majority of autonomous driving researchers.

We collect training data at a frequency of 10Hz, accumulating 30 hours of driving data from four distinct training towns (Town01, Town03, Town04, Town06) under four different weather conditions (ClearNoon, WetNoon, HardRainNoon, ClearSunset). At each timestep, we store the data tuple $(s_t, a_t, r_t)$, whose compositions will be introduced in detail in the next paragraphs.

**Reward:** The symbol $r_t \in \mathbb{R}$ denotes the reward returned by the environment at the timestep $t$. Typically, our reward design is revised from Roach (Zhang et al., 2021), encompassing factors such as speed, safety, and comfort.

$$r = r_{\mathrm{speed}} + r_{\mathrm{position}} + r_{\mathrm{rotation}} + r_{\mathrm{action}} + r_{\mathrm{terminal}}. \tag{24}$$

Among them, $r_{\mathrm{speed}} = 1 - \frac{|v - v_{\mathrm{desired}}|}{v_{\mathrm{max}}}$ represents the reward of approaching the target speed; $r_{\mathrm{position}} = -0.5\Delta_p$ represents the reward obtained from driving in the correct position, where $\Delta_p$ is the lateral distance between the ego vehicle and the centerline of the target route; $r_{\mathrm{rotation}} = -\Delta_r$ represents the reward obtained from driving in the correct orientation, where $\Delta_r$ is the angle difference between the ego vehicle and the centerline of the target route; $r_{\mathrm{action}}$ incurs a penalty of -0.1 when the steering in current timestep differs from the previous step by more than 0.01, promoting smooth and comfortable driving; $r_{\mathrm{terminal}}$ is 10 when reaching the destination, -10 when encountering a collision or violation of traffic rules, to encourage driving safely.

**Safety constraints:** In addition to the reward setting, the Constrained Penalized Q-learning (CPQ) (Xu et al., 2022) algorithm also needs a constraint definition for training. Specifically, we adopt the same setting as in (Li et al., 2022b), where each time of collision or traffic rule violation incurs a penalty of 1.0. The constraint limit for CPQ is 1.0 (i.e. ensures no safety issue at each ride).

In order to avoid memory overflow in the sequence model, both the action space and state space have been specially designed. Specifically, we adopt a similar setting to SPLT (Villaflor et al., 2022).

**Action:** The action $a \in \mathbb{R}^2$ consists of two values, representing the target angle and target velocity that will be fed into two separate PID controllers (Johnson & Moradi, 2005).

**State:** The observed state $s \in \mathbb{R}^{37}$ includes the following components: (i) ego vehicle velocity (ii) relative distance and velocity to the leading vehicle, which will be set to the default maximum value if no leading vehicle is perceived in the horizon (iii) relative distance and velocity to the pedestrian ahead, which will be set to default maximum values if no pedestrian is perceived in the horizon (iv) relative velocities and distances to vehicles in four cardinal directions (front, rear, left, right), also set to default maximum values if no vehicles are perceived in the horizon (v) distance to the traffic light ahead, set to default maximum value if no red light ahead (vi) distance to the stop sign ahead, set to default maximum value if no stop sign ahead (vii) relative angle difference w.r.t. the centerline of the target route (viii) distance to the centerline of the target route (ix) relative positions of the next 10 target waypoints.

# E    IMPLEMENTATION DETAILS

In this section, we introduce the implementation details, including our training process, evaluation process, and hyperparameters. Specifically, UNREST and all the baselines are implemented with Python 3.7 and PyTorch 1.13.1. Besides, all training processes are run on a NVIDIA A10 while inference is conducted on one NVIDIA 3090 GPU for fair comparison.

## E.1    TRAINING DETAILS

**Training Process:** For all training processes, We employ an AdamW optimizer with a learning rate of $10^{-4}$, and a consistent batch size of 256. For the training of sequence models, a history length of 10 is carefully selected to be sampled each time, which corresponds to a real-world duration of 1 second. To determine the number of epochs for training, a dynamic value is assigned based on the dataset's size, defined by the following formula: $n_{\text{epochs}} = \text{int}\left(\frac{1e6}{\text{len(dataset)}} \times n_{\text{ref}}\right)$, where $n_{\text{ref}}$ represents a reference epoch number that can be tuned.

**Details for segmentation:** As discussed in Sec. 4.3, we train two return transformers for sequence segmentation. In practical implementation, to accurately quantify environmental uncertainty, we employ the ensemble method (Lakshminarayanan et al., 2017a) to train each return transformer. Using Eq. 11, we recalculate the variance and expectation of the returns. Then the sequences are segmented into certain and uncertain parts w.r.t. estimated uncertainties according to the principles we introduce in Sec. 4.3.

In our experiments, we discover that 'uncertain states' tend to occur consecutively. To ensure meaningful segmentation, we enforce a minimum 'uncertain part' length of $c = 20$ frames and select the uncertainty threshold $\epsilon = 3$ according to Fig. 5(a). Under these two hyperparameter settings, we find the segmented sequences often correspond to the completion of a significant driving task.

**Details for return-span embedding:** We introduce the return-span embedding $f_\theta^h(h_t)$ to enhance the interpretability of truncated returns as conditions. Specifically, we explore two variants to incorporate this embedding. The first variant is to add the return-span embedding to the return embedding, as stated in the main text. The second variant considers treating $f_\theta^h(h_t)$ as a distinct token input to the sequence model. However, we observe that the latter approach results in increased memory consumption without yielding improvements in performance (Tab. 8 and Tab. 9).

**Details for global return embedding:** The approach for obtaining the global return embedding differs from other embeddings due to its high uncertainty. Typically, instead of utilizing parameterized networks, we opt for a coarse uniform discretization approach, resulting in a corresponding one-hot vector (specifically, a 50-dimensional vector in our implementation). This strategy enables us to provide global guidance while mitigating the influence of its uncertainty on the training process.

**Details for decision models:** In the practical implementation, following DT (Chen et al., 2021), we directly predict the value of the action instead of its distribution. Specifically, we only retain the mean

Table 4: Hyperparameters for training return transformers for Sequence Segmentation.

| Hyperparameters | Value | Hyperparameters | Value |
|---|---|---|---|
| # Transformer layers | 4 | # Transformer heads | 8 |
| Embedding dimension | 128 | Batch size $B$ | 256 |
| Sampled Sequence length | 10 | Discount $\gamma$ | 0.95 |
| Learning rate | 1e-4 | Dropout | 0.1 |
| Optimizer | AdamW | Weight decay | False |
| Ensemble size $K$ | 5 | Data mask probability | 0.6 |
| # Reference epoch $n_{\text{ref}}$ | 50 | Transformer Activation | GELU |

$\mu_\theta$ and assume unit variance. Consequently, the loss in Eq. 9 can be reformulated as the mean square loss, resembling the learning objective in DT.

### E.2 EVALUATION DETAILS

**Main setting:** To comprehensively evaluate our models' planning capability, we select two distinct driving scenarios from CARLA: the training town under training weather conditions and the new town under new weather conditions. Typically, the training town is chosen as Town03, the most complex town among the training dataset, and the training weathers are 'WetNoon' and 'ClearSunset'. The new town is Town05, the most complex town in the rest of the training set, and the new weathers are 'SoftRainSunset' and 'WetSunset'. For each tested scenario, we carefully set up the vehicles and driving routes w.r.t. the specifications of CARLA Leaderboard (team, 2020). Moreover, we conduct experiments with three different seeds (2022, 2023, 2024) to facilitate the calculation of mean values and variances.

**Details for dynamic uncertainty estimation:** As we have stated in the main text, we employ dynamic uncertainty estimation at inference time to enable cautious planning. Typically, we implement more variants (results are shown in Tab. 8 and Tab. 9) other than KD-Tree (Redmond & Heneghan, 2007) for test-time uncertainty estimation:

- **Network prediction:** The first approach entails training a neural network to forecast environmental uncertainty. Leveraging the uncertainty values computed by the aforementioned return transformers and employing the provided threshold, we assign binary class labels (*certain* and *uncertain*) to each state in the training dataset. Subsequently, these labels are utilized to train an uncertainty network.
- **Heuristic:** The heuristic method turns out to be a straightforward and intuitive alternative. Specifically, we observe that specific dimensions within received states have a strong correlation with uncertainties (e.g. distance to traffic light ahead). Thus, we directly compare the values of these particular dimensions to the preset uncertainty threshold to determine whether the state is uncertain.
- **KD-tree:** Ultimately, our chosen methodology employs a KD-tree for uncertainty estimation. Initially, we construct a KD-tree that stores both the states and the uncertainties computed by the trained return transformers. As test time, we sample the nearest neighbors of the current state from the tree and compute the mean value of neighbors as the current state's uncertainty.

### E.3 HYPERPARAMETERS

We select several well-established approaches to serve as our comparative baselines. Specifically, we include Behavior Cloning (BC), Implicit Q-Learning (IQL) (Kostrikov et al., 2021), as well as prominent sequence models such as Decision Transformer (DT) (Chen et al., 2021), Trajectory Transformer (TT) (Janner et al., 2021), SPLT (Villaflor et al., 2022), and ESPER (Paster et al., 2022). To ensure a fair comparison, for all these baselines, we adopt the default hyperparameter settings from the repository `https://github.com/avillaflor/SPLT-transformer`. For Monotonic Advantage Re-Weighted Imitation Learning (MARWIL) (Wang et al., 2018) and Conservative Q-Learning (CQL) (Kumar et al., 2020), we adopt the default hyperparameter from the Ray RLlib `https://docs.ray.io/en/latest/rllib/index.html`. Finally for the safe offline RL baseline Constraints-Penalized Q-learning (CPQ) (Xu et al., 2022), we employ the default hyperparameters from the repository `https://github.com/liuzuxin/OSRL`.

The full list of UNREST's hyperparameters can be found in Tab. 4, Tab. 5, and Tab. 6.

Table 5: Hyperparameters for training UNREST decision models.

| Hyperparameters | Value | Hyperparameters | Value |
|---|---|---|---|
| Uncertainty threshold $\epsilon$ | 3.0 | Min 'uncertain part' length $c$ | 20 |
| # Transformer layers | 4 | # Transformer heads | 8 |
| Embedding dimension | 128 | Batch size $B$ | 256 |
| Sampled Sequence length | 10 | Discount $\gamma$ | 1.0 |
| Optimizer | AdamW | Weight decay | False |
| Learning rate | 1e-4 | Dropout | 0.1 |
| # Reference epoch $n_{ref}$ | 200 | Transformer Activation | GELU |
| Global return dimension | 50 | Action Activation | Tanh |

Table 6: Hyperparameters for UNREST's inference process.

| Hyperparameters | Value | Hyperparameters | Value |
|---|---|---|---|
| Max history length | 5 | Uncertainty threshold $\epsilon$ | 3.0 |
| Return Horizon $H$ | 100 | Upper Percentile $\eta$ | 0.7 |
| Deterministic sample | True | # KD-Tree neighbor | 5 |

# F    MORE EXPERIMENTAL RESULTS

In this section, we supplement more experimental results including the visualizations, complexity comparisons, sensitivity analysis of hyperparameters, and the results for more UNREST variants.

## F.1    VISUALIZATIONS

**Uncertainty visualizations:** We present more visualizations of UNREST's uncertainty estimation in Fig. 5 to validate its interpretability. Initially, we visualize the uncertainty distribution in Fig. 5(a). As we can see the majority of uncertainty associated with state transitions falls within the range of 0 to $10^{-4}$, aligning with intuitive expectations. In the context of autonomous driving environments with sparse vehicle presence and a limited number of traffic signals, the stochasticity of environment transitions tends to be relatively low. The uncertainty threshold $\epsilon$ is accordingly chosen as 3.0 to cover corner cases with large uncertainties.

Then we seek uncertain transition states within the training dataset and visualize two typical scenarios. Fig. 5(b) and Fig. 5(d) correspond to a turning scenario, while Fig. 5(c) and Fig. 5(e) depict a scenario of passing through a red light. By associating the temporal steps with the uncertainty curves, it can be observed that as the ego-vehicle approaches the turning point, the uncertainty gradually increases since the ego-vehicle cannot forecast the traffic conditions in the lane after turning. After the turning behavior is almost completed, the uncertainty decreases below the threshold again. In the scenario of waiting at a red light, it can be observed that the uncertainty rapidly increases as the ego-vehicle approaches the red light due to the uncertainty in color changes of the traffic signal. However, after the ego-vehicle comes to a stop in front of the red light for approximately 15 frames, the uncertainty decreases sharply below the threshold. These visualization results effectively demonstrate the interpretability of our uncertainty measurement approach.

**Uncertainty Calibration:** To provide a more intuitive impression of UNREST's effectiveness, we present in Fig. 6 the return distribution calibration results. Typically, the figure illustrates that using an ensemble of return transformers can significantly better calibrate the return distribution than using a single model (closer to the ground truth, with tight uncertainty bands), where it achieves smaller (better) results on two widely adopted calibration metrics: RMSE and NLL (Lakshminarayanan et al., 2017b) than the single return transformer variant. Notably, the uncertainty in the figure denotes the variance of the distribution, *different from the so-called environmental uncertainty we used in the main text*, that reflects the impact of environmental transitions (through conditional mutual information). The predicted return distribution has corresponding ground truth and thus can be directly calibrated like what we do in the figure. However, the impact of the environment has no ground truth value and can only be interpreted through visualizations like Fig. 4 and Fig. 5.

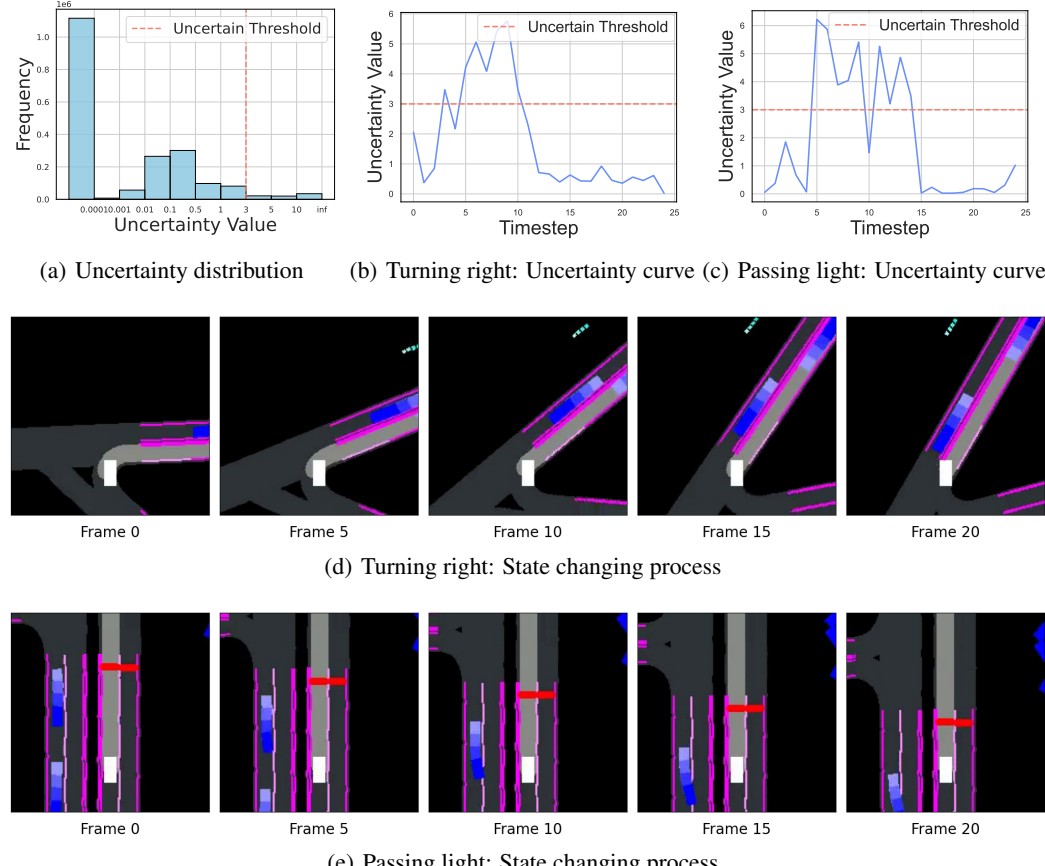

(a) Uncertainty distribution     (b) Turning right: Uncertainty curve    (c) Passing light: Uncertainty curve

(d) Turning right: State changing process

(e) Passing light: State changing process

Figure 5: More visualizations of UNREST's uncertainty estimation results.

Table 7: Comparison results of training/inference time ($10^{-2}$s) and training GPU memory (GB) between different sequence models. The training time is calculated w.r.t. multiple iterations, while the inference time is calculated w.r.t. multiple rollout steps. The GPU usage is a fixed value for one training model.

| Metric | BC | DT (Chen et al., 2021) | TT (Janner et al., 2021) | SPLT (Villaflor et al., 2022) | UNREST |
|---|---|---|---|---|---|
| GPU usage | 1.00 | 1.46 | 23.3 | 1.96 | 1.48 |
| Training time | $0.83 \pm 0.13$ | $0.93 \pm 0.13$ | $32.7 \pm 0.2$ | $4.20 \pm 0.1$ | $1.47 \pm 0.12$ |
| Inference time | $0.15 \pm 0.01$ | $0.16 \pm 0.02$ | $25.6 \pm 0.7$ | $0.89 \pm 0.02$ | $0.24 \pm 0.03$ |

**Sequence length distributions:** We present different segmented sequence length distributions in Fig. 7. Analyzing the figures, it becomes evident that most uncertain sequences tend to possess lengths that are in close proximity to the step-length $c$, while certain sequences are almost uniformly distributed w.r.t. their lengths.

## F.2 COMPLEXITY COMPARISONS

We provide empirical results of space/time usage as shown in Tab. 7. In terms of GPU usage, UNREST demonstrates comparable levels to DT, while significantly lower utilization compared to SPLT and TT. Regarding training and inference time, UNREST consumes slightly more time than DT, yet notably faster than TT and SPLT. These results signify that UNREST achieves superior performance while requiring relatively modest computational resources.

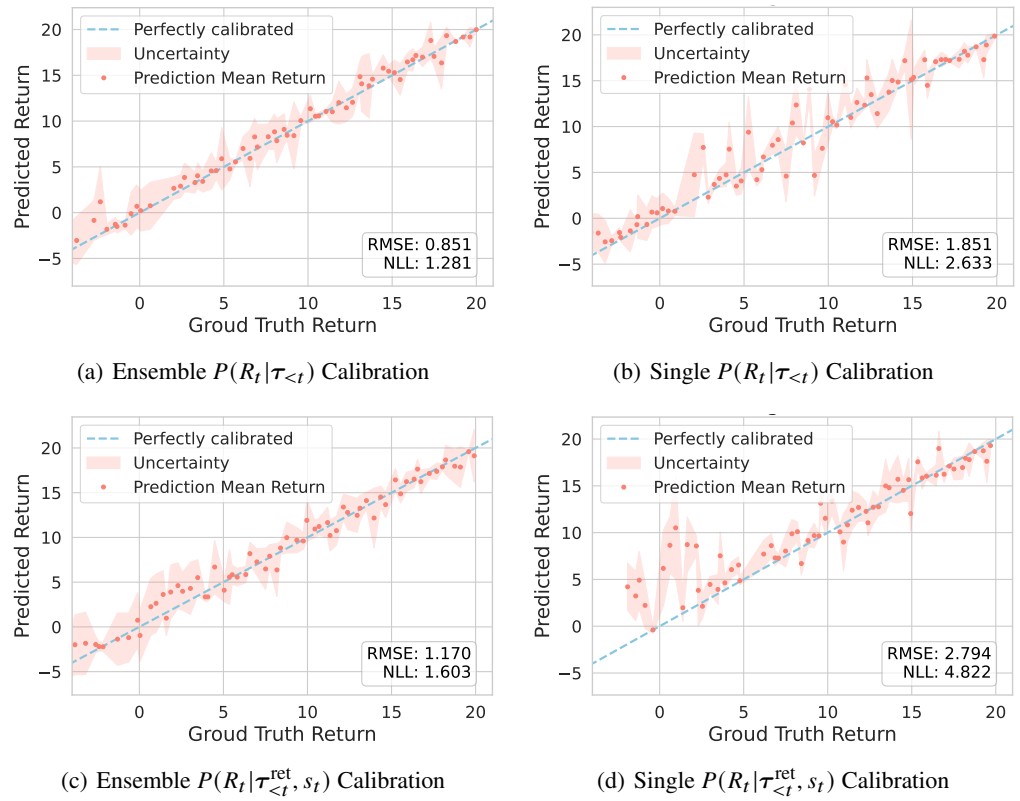

(a) Ensemble $P(R_t|\tau_{<t})$ Calibration

(b) Single $P(R_t|\tau_{<t})$ Calibration

(c) Ensemble $P(R_t|\tau_{<t}^{\text{ret}}, s_t)$ Calibration

(d) Single $P(R_t|\tau_{<t}^{\text{ret}}, s_t)$ Calibration

Figure 6: Calibration results of return distribution using ensemble models are obviously better than that using a single model. We use the standard variance of the networks' predictions as an approximate indicator of uncertainty. The blue line signifies the ground truth, while the red dots denote the predicted mean returns. The areas shaded in orange represent the predicted mean coupled with their respective standard deviations.

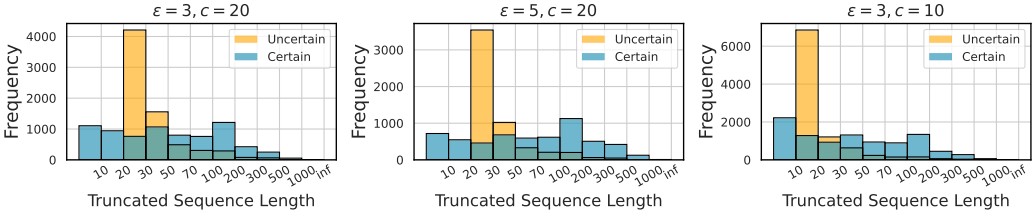

Figure 7: Segmented sequence length distributions with different threshold $\epsilon$ and step-length $c$.

## F.3 SENSITIVITY ANALYSIS

**Impact of return horizon $H$:** We investigate the influence of the return horizon length, denoted as $H$, on the performance of UNREST. The experimental results are depicted in Fig. 8, where the return horizon varies from 50 to 1000. It is evident that a relatively modest value of $H$ (e.g., $H$=100) yields the most favorable performance, while both excessively small and excessively large values of $H$ exhibit noticeable performance deterioration. These findings align with our underlying assumption that shorter sequences can alleviate uncertainty, while excessively short sequences may give rise to shortsightedness, thereby compromising performance.

**Impact of uncertainty threshold $\epsilon$:** We next investigate the impact of the uncertainty threshold $\epsilon$ and present the results in Fig. 8 where we increase $\epsilon$ from 0.5 to 10. As is shown in the figure, a moderate value of $\epsilon$, namely $\epsilon = 3$ contributes to the best performance, while both smaller and larger

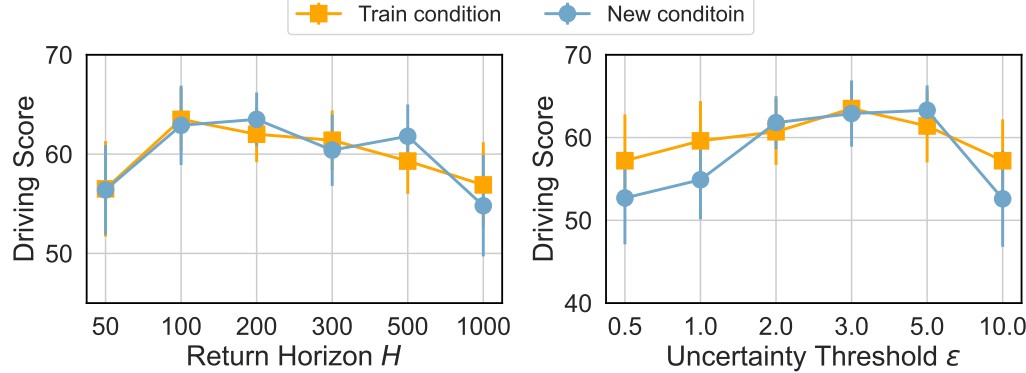

Figure 8: The sensitivity analysis of hyper-parameters $H$ and $\epsilon$.

Table 8: Driving performance of more UNREST variants on train town and train weather condition.

| Planner | Driving Score↑ | Success Rate↑ | Route Comp.↑ | Infrac. Score↑ | Norm. Reward↑ |
|---|---|---|---|---|---|
| Tokened ret-span emb. | 62.8 ± 3.6 | 56.2 ± 4.8 | 81.1 ± 4.7 | 66.5 ± 2.5 | 0.62 ± 0.03 |
| Fixed horizon seg. | 59.2 ± 2.6 | 44.7 ± 3.2 | 78.3 ± 4.8 | 64.5 ± 3.0 | 0.66 ± 0.02 |
| Separate BC model | 61.0 ± 4.1 | 51.7 ± 5.3 | 82.8 ± 3.7 | 64.6 ± 2.4 | 0.68 ± 0.04 |
| Heuristic uncertainty | 62.1 ± 2.4 | 28.8 ± 7.4 | 43.8 ± 6.9 | 72.6 ± 4.3 | 0.47 ± 0.03 |
| Predicted uncertainty | 60.7 ± 2.7 | 50.0 ± 4.4 | 63.6 ± 5.2 | 65.8 ± 3.4 | 0.61 ± 0.02 |
| Reweighted BC | 64.2 ± 2.4 | 57.5 ± 3.3 | 85.5 ± 2.7 | 71.4 ± 2.5 | 0.68 ± 0.03 |
| Original model | 63.5 ± 3.2 | 54.5 ± 7.0 | 83.8 ± 3.1 | 70.2 ± 2.8 | 0.64 ± 0.04 |

Table 9: Driving performance of more UNREST variants on new town and new weather conditions.

| Planner | Driving Score↑ | Success Rate↑ | Route Comp.↑ | Infrac. Score↑ | Norm. Reward↑ |
|---|---|---|---|---|---|
| Tokened ret-span emb. | 62.0 ± 4.9 | 56.8 ± 3.0 | 92.0 ± 5.4 | 60.3 ± 3.3 | 0.64 ± 0.04 |
| Fixed horizon seg. | 58.5 ± 3.6 | 46.7 ± 6.2 | 80.3 ± 2.8 | 64.7 ± 2.8 | 0.64 ± 0.02 |
| Separate BC model | 59.3 ± 3.4 | 63.8 ± 4.3 | 90.0 ± 5.5 | 59.3 ± 4.4 | 0.68 ± 0.05 |
| Heuristic uncertainty | 56.7 ± 2.6 | 40.0 ± 7.3 | 70.0 ± 6.2 | 72.5 ± 3.0 | 0.52 ± 0.02 |
| Predicted uncertainty | 59.6 ± 3.4 | 55.6 ± 5.5 | 88.0 ± 3.7 | 58.6 ± 4.1 | 0.64 ± 0.04 |
| Reweighted BC | 63.8 ± 2.3 | 59.3 ± 4.7 | 91.0 ± 5.0 | 65.1 ± 3.7 | 0.67 ± 0.02 |
| Original model | 62.9 ± 4.0 | 57.5 ± 5.4 | 90.0 ± 6.0 | 62.9 ± 3.8 | 0.65 ± 0.03 |

values of $\epsilon$ exhibit obvious performance degradation. This behavior can be attributed to the fact that the majority of uncertainty values fall within the range of 0 to $10^{-4}$. Consequently, excessively small values may incorrectly identify 'certain states' as uncertain, while excessively large values may overlook important 'uncertain states'.

## F.4 MORE UNREST VARIANTS

Tab. 8 and Tab. 9 record the detailed performance of more UNREST variants in both training and new scenarios. It can be seen from the results that introducing return-span embedding as a separate token does not lead to a significant improvement but will increase memory usage. Fixed horizon segmentation means segmenting the sequences at a fixed timestep 100. However, it ignores environmental uncertainties and may lead to shortsighted problems. Therefore, it's reasonable that it performs notably worse than our original model. The Separate BC model means separately training an offline RL model in 'certain parts' and a BC model in 'uncertain parts'. While achieving the highest normalized rewards, there is still a significant gap in its driving score compared with the original model, which shows that the joint training of two models can benefit the planning process. Utilizing a heuristic method to estimate uncertainties at inference time obtains the best infraction score, which is apparent since it is designed to promote cautious behavior in every possible uncertain scenario. However, this method is over-pessimistic and overlooks certain corner cases, ultimately

Table 10: Comparison to SOTA baselines on the standard D4RL Mujoco locomotion-v2 domain. For BC, MBOP, CQL, DT, TT, IQL, and SPLT we use the results reported from the SPLT paper (Villaflor et al., 2022). For ESPER, we report the results from their paper (Paster et al., 2022). We report the mean and std for our method over 3 seeds with 10 trajectories for each seed. 'Stochastic' denotes customized stochastic environments according to (Yang et al., 2022). The symbol '-' is used to indicate the omission of measurement results that are deemed unimportant.

| Dataset | Environment | BC | MBOP | CQL | DT | TT | IQL | SPLT | ESPER | UNREST(ours) |
|---|---|---|---|---|---|---|---|---|---|---|
| Med-Expert | HalfCheetah | 59.9 | 105.9 | 91.6 | 86.8 | 95.0±0.2 | 86.7 | 91.8±0.5 | 66.95±11.13 | 91.9±0.8 |
| Med-Expert | Hopper | 79.6 | 55.1 | 105.4 | 107.6 | 110.0±2.7 | 91.5 | 104.8±2.6 | 89.95±13.91 | 93.5±5.4 |
| Med-Expert | Walker2d | 36.6 | 70.2 | 108.8 | 108.1 | 101.9±6.8 | 109.6 | 108.6±1.1 | 106.87±1.26 | 105.74±4.43 |
| Medium | HalfCheetah | 43.1 | 44.6 | 44.0 | 42.6 | 46.9±0.4 | 47.4 | 44.3±0.7 | 42.31±0.08 | 44.5±0.9 |
| Medium | Hopper | 63.9 | 48.8 | 58.5 | 67.6 | 61.1±3.6 | 66.3 | 53.4±6.5 | 50.57±3.43 | 79.84±4.18 |
| Medium | Walker2d | 77.3 | 41.0 | 72.5 | 74.0 | 79.0±2.8 | 78.3 | 77.9±0.3 | 69.8±1.2 | 70.6±3.9 |
| Med-Replay | HalfCheetah | 4.3 | 42.3 | 45.5 | 36.6 | 41.9±2.5 | 44.2 | 42.7±0.3 | 35.9±2.0 | 39.4±0.5 |
| Med-Replay | Hopper | 27.6 | 12.4 | 95.0 | 82.7 | 91.5±3.6 | 94.7 | 75.0±23.8 | 50.2±16.1 | 65.7±21.9 |
| Med-Replay | Walker2d | 36.9 | 9.7 | 77.2 | 66.6 | 82.6±6.9 | 73.9 | 57.7±4.7 | 65.5±8.1 | 66.3±6.4 |
| **Average** | | 47.7 | 47.8 | 77.6 | 74.7 | 78.9 | 76.9 | 72.9 | 64.2 | 73.0 |
| Med-Stochastic | HalfCheetah | - | - | - | 75.8±1.2 | 83.7±2.8 | - | 88.6±2.2 | 77.4±5.6 | **90.5 ± 3.2** |
| Med-Stochastic | Hopper | - | - | - | 89.4±3.3 | 93.4±3.6 | - | 94.8±2.7 | 92.5±6.9 | **97.4 ± 4.4** |

leading to the lowest driving score. Adopting a network to predict uncertainty at inference time yields better performance than the heuristic method, but it can often misclassify certain/uncertain states and still performs worse than the original model. Generally, the original UNREST model achieves the best driving scores in both training and new driving scenarios, which is the reason why we choose it as our ultimate planning framework. Finally, we also study a Reweighted BC variant, which additionally fit state value functions like that in MARWIL (Wang et al., 2018) and conduct advantage reweighted behavior cloning at uncertain parts. Since the offline data collected by AutoPilot can be suboptimal, we find reweighted behavior cloning can effectively down-weight those suboptimal actions and improve UNREST's driving scores at both training and new scenarios. However, due to its significant increase in model complexity, we have refrained from incorporating this technique into the final model.

## F.5 RESULTS ON D4RL

For completeness, we evaluate our method on D4RL Mujoco tasks, though they are generally deterministic tasks. Specifically, we find our UNREST is generally competitive with SOTA baselines on these tasks, as shown in Tab. 10. The reason for his overall underperformance compared to DT lies in the unnecessarily forced sequence segmentation in such deterministic tasks, where it is more appropriate to retain global returns in order to extend the horizon length considered during planning. We also test UNREST in two customized stochastic D4RL environments from (Yang et al., 2022), where UNREST achieves the highest return among the models.

