# OpenReview forum: "Uncertainty-Aware Decision Transformer for Stochastic Driving Environments"
_ICLR.cc/2024/Conference — Submitted to ICLR 2024_

### Official Review · Reviewer_nCLY · 2023-10-30

**Soundness:** 2 fair
**Presentation:** 2 fair
**Contribution:** 2 fair
**Rating:** 3
**Confidence:** 4

**Summary:**

The paper addresses the challenge of utilizing offline Reinforcement Learning (RL) for autonomous driving tasks, especially in stochastic environments where unpredictable events can occur. Traditional methods, influenced by the successes of the Transformer architecture, have reimagined offline RL as sequence modeling. However, these methods, known as decision transformers, tend to make overly optimistic assumptions in unpredictable environments. They often incorrectly believe that a particular action, once successful, will always lead to the desired outcome. This is problematic because in real-world driving, identical actions can lead to vastly different outcomes based on unpredictable factors like the behavior of other vehicles. The crux of the issue lies in differentiating the outcomes resulting directly from the agent's actions versus those emerging from the unpredictable environment. Current attempts to solve this have primarily focused on creating state transition models, but these are intricate and demand highly representative environment models, which are hard to achieve for intricate driving tasks. To combat this, the paper introduces the UNcertainty-awaRE decision Transformer (UNREST). The core idea is to customize decision transformers for unpredictable driving scenarios without adding more complexity. UNREST achieves this by estimating state uncertainties via the mutual information between transitions and returns.

**Strengths:**

1. Using the offline RL framework to solve driving tasks is a promising and interesting direction as online training is not feasible due to the safety issue. This paper modifies a state-of-the-art framework, i.e., a decision transformer to achieve better performance considering the uncertainty of the environment.

2. The evaluation part is comprehensive, including many baseline methods and evaluation metrics. There are also many ablation studies that investigate the influence of different modules of the model.

**Weaknesses:**

1. Where does the uncertainty in autonomous driving come from? My understanding of uncertainty is that with the same (s, a) pairs, the environment returns different rewards. Figure 1 tries to provide an example but a lot of information is missing. For example, what is the purpose of the traffic light? What does the x-axis label “return” mean for task 1 and task 2? Do the authors mean reward to go?
2. The motivating example of the uncertainty in the driving scenario seems to involve the tradeoff between reward (reaching point) and cost (collision). In my opinion, tuning the weights in reward functions or some other parameters could dramatically influence the “uncertainty” investigated in this paper. Basically, it will lead to either aggressive or conservative driving policy. However, cost or constraint is only discussed in the appendix. Therefore, I doubt that the proposed transformer model can correctly estimate the uncertainty.
3. It seems that one important problem investigated in this paper is “Uncertainty Accumulation”, which is reasonable since the uncertainty increases as the trajectories accumulate more steps. However, why do we care about the accumulation of uncertainty in driving tasks? A normal driving algorithm just needs to make decisions based on current uncertainty and historical information. I guess one reason could be the DT framework uses reward-to-go as a condition. If this is the case, this paper may not be applicable to other frameworks.
4. A lot of important information is missing in figures, experiments, and conclusions.

**Questions:**

1. After checking Property 1 and Property 2, I still cannot fully understand Figure 1. For example, what does “normalized groud truth” mean (I assume “groud” is a typo of “ground”)? I highly suggest the authors provide a simplified and intuitive example in the first figure without adding any distracting information. The detailed explanation can be put in later sections.
2. I noticed that the offline data is collected in Carla with Autopilot. How to show that the data contains uncertainty? Should the uncertainty come from the behavior of surrounding vehicles?
3. Missing information in Figure 2. What does h represent? Why does h start from 1 but end with 80, 79?
4. Information missing in Figure 3. What does the light blue rectangle mean? Which one is the ego vehicle? What does failure mean in Figure 3(a)? It seems that the lane change is not available as there is another car in the target lane. What’s the difference between Figure 3(a) and Figure 3(b)?

---

> ### Author Response · Authors · 2023-11-17
> **Response to Reviewer nCLY (1/3).**
>
> Thank you for your time and valuable advice on improving our paper. We are pleased that you acknowledged the motivation and experiments of our paper. Here we provide answers to all of your questions one-by-one as follows (also see the revised paper in the updated pdf):
>
> > ***Q1: Where does the uncertainty in autonomous driving come from? There are some missing information in Figure 1.***
>
> **R1:** Thank you for highlighting the problem, which is crucial for understanding our paper. Specifically, as discussed in the introduction, **uncertainty in the autonomous driving environment may arise from the hidden states of other vehicles and partial observability.** In other words, the transition $(s,a\rightarrow s')$ along with the reward $r$ is inpredictable (different possible outcome $s'$ and $r$) because the surrounding vehicles can behave stochastically and some information is not observable.
>
> Based on your suggestion, we have made revisions to Figure 1 and its corresponding description. Specifically, we remove the irrelevant traffic lights from the figure. Besides, we clarify that the term 'return' expresses exactly the same meaning as 'reward-to-go' in reinforcement learning, about the cumulative rewards in the episode. Regarding Figure 1(a), it represents a driving scenario composed of a sequence of driving actions, starting with going straight and followed by a right turn. In this scenario, the uncertainty in the environment is relatively low during the straight driving process, while the variance in the distribution of rewards increases during the right turn due to the uncertain behavior of other vehicles in the environment. Therefore, we propose to condition solely on the truncated return from Task I (going straight) rather than Task I & II to mitigate the influence of environmental stochasticity (less stochastic timesteps, lower return variance) for DT training, which still preserves rewards over sufficient timesteps for the optimization of current actions.

---

> ### Author Response · Authors · 2023-11-17
> **Response to Reviewer nCLY (2/3).**
>
> > ***Q2: The motivating example involves the tradeoff between reward and cost. The proposed transformer model may not correctly assess the uncertainty when tuning weights in reward functions.***
>
> **R2:** Thank you for pointing out the problem of our reward design. I believe it is necessary to clarify that **all the reinforcement learning algorithms we employ do not exhibit the so-called trade-offs. Instead, their primary objective is to purely seek strategies that maximize rewards.**
>
> As you said, we have discussed the reward and cost compositions of our dataset in Appendix D. **However, for the RL algorithms, the cost is integrated in the reward function as penalization for dangerous behaviors.** Otherwise, the RL agent may frequently conduct aggressive actions that may raise safety issues. So for the example driving scenario in Figure 1(a), the ego-vehicle will learn to yield to surrounding vehicles, thereby avoiding significant negative rewards resulting from collisions.
>
> Furthermore, as you mentioned, adjusting the weights in the reward function will inevitably lead to variations in the estimation of uncertainties. However, **we do not consider this to be a problem because the essence of the effectiveness of RL lies in the design of a well-defined reward function.** While adjusting the weights in the reward function may harm uncertainty estimation, the more significant concern is that arbitrary adjustments can lead to suboptimal policy learning by the RL algorithm itself. For our approach, since we are identifying uncertainty in returns (rewards), when the reward function can serve as an effective conditioned target for DT policy learning, UNREST will also be capable of effective segmenting sequences for DTs.
>   | Planner | CostWeight=0.1  | CostWeight=5.0 | CostWeight=10.0 | CostWeight=50.0  |
>   | ------- | --------------------------- | --------------------------- | --------------------------- | --------------------------- |
>   | DT  | $48.2\pm1.6$ ($40.4\pm2.4$) | $54.4\pm2.8$ ($47.4\pm1.7$) | $55.2\pm2.0$ ($47.6\pm1.2$) | $52.6\pm3.0$ ($46.5\pm2.3$) |
>   | UNREST | $48.4\pm2.4$ ($41.3\pm1.5$) | $63.0\pm2.2$ ($61.9\pm3.0$) | $63.5\pm3.2$ ($62.9\pm4.0$) | $62.8\pm2.5$ ($61.5\pm1.9$) |
>
> The above table records our new testing driving scores on train and new (in brackets) scenarios w.r.t. different cost weights in reward functions. As can be seen from the table, both excessively high and excessively low weights can result in the inability of the DT to train effective policies. However, the introduction of uncertainty estimation in UNREST consistently brings improvements compared to DT across different weight settings.
>
> > ***Q3: Why do we care about the accumulation of uncertainty in driving tasks? If it can be attributed to DTs, this paper may not be applicable to other frameworks.***
>
> **R3:** Thank you for pointing out the problem. Exactly as you said, the uncertainty accumulation is a normal characteristic within the context of autonomous driving environments. **However, our attention to this property is not solely due to the DT framework.** Typically, it's natural to consider uncertanity accumulation for DTs, since the conditioned reward-to-go includes the uncertainties at each time step from the current state to the end of the sequence. UNREST, however, addresses and reduces this conditioning uncertainty by segmenting the sequences, enabling more effective policy learning in stochastic environments.
>
> **But for other RL algorithms, uncertainty accumulation is also a significant factor to take into consideration since the monte-carlo/trajectory statistics are fundamental factors in RL**, which utilize simulated trajectory return to update the value functions, or directly select a better action. In these simulated trajectories, the cumulated uncertainty can lead to large variance during backup/forward searching. **Our uncertainty estimation is applicable in these frameworks by accurately identifying uncertain timesteps and branching out to perform additional trajectory rollouts at those points.** This process helps improve the accuracy of the monte-carlo estimation results，which suggests the general use cases of our uncertainty estimation. For example, [1, 2] propose to learn transition models for monte-carlo simulation. Due to the significant bias associated with single-step rollouts, they have all opted for multi-step beam search to find the most effective and safe actions. By combining our uncertainty estimation method, we can effectively identify specific uncertain states that require branching and rollout. This can greatly helps reduce the number of simulations and improves planning efficiency.

---

> ### Author Response · Authors · 2023-11-17
> **Response to Reviewer nCLY (3/3).**
>
> > ***Q4: Missing information through the paper: (1) A simplified and intuitive example is preferred in Figure 1. (2) How to show the data collected from CARLA contains uncertainty (e.g. from the uncertain behavior of surrounding vehicles)? (3) Missing information in Figure 2. (4) Missing information in Figure 3.***
>
> **R4:** Thank you for pointing out our oversights in writing. We have made necessary modifications to the corresponding content in our updated version of paper (highlight in green and blue) and here we provide individual responses to all of your questions.
>
> 1. As discussed in R1, we have made revisions to Figure 1 and its corresponding description. Regarding Figure 1(a), it represents a driving scenario composed of a sequence of driving actions, starting with going straight (low uncertainty) and followed by a right turn (high uncertainty). We also remove the irrelevant traffic light from the figure and clarify the task and return distribution contents in the figure. For Figure 1(b) and Figure 1(c), we have modified their x-axis labels to provide greater precision, specifically as, 'Normalized Return (Ground Truth)' and 'Timestep', respectively. Typically, the x-axis in Figure (b) represents the ground truth return in original trajectories. More detailed explanations are expressed in the main context of the newly updated paper.
> 2. Yes, our training data is indeed collected from CARLA. And as we have discussed in R1, the uncertainty can arise from the stochastic behavior of surrounding vehicles and the partial observability. Two respective examples can be referred in Figure 4 in the paper. Typically, in Figure 4(a) we observe an initial increase in uncertainty as the ego-vehicle enters the lane to follow another vehicle, owing to the lack of knowledge about the other vehicle's behavior. This uncertainty gradually decreases back below the threshold when the vehicle stabilizes in the following state. In Figure 4(b), while approaching the green light, the partial observability about the light state change causes the uncertainty to rise quickly. After the vehicle has moved away from the traffic light, the uncertainty immediately drops back below the threshold.
> 3. The meaning of $h$ is expressed as 'return-span token' in Figure 2, which is further explained below Equation (6). Specifically, the return-span $h_t$ denotes the timestep count involved in the truncated return $R^h_t$. As we discussed in Section 4.3, for states in certain parts, $h$ is set to the number of timesteps to next segmentation position to include the maximum duration that do not contain any uncertain steps; for states in uncertain parts, $h$ is set as empty for meaningless conditions, to conduct behavior cloning. So in Figure 2, we start with $h=1$, which is the final timestep in a certain part. Afterwards, the state is identified as uncertain and an uncertain part (orange background) begins with empty $h$. Finally, another certain part with 80 timesteps starts, with $h=80$, then sequentially decreases to $h=79, 78, ...$ until another uncertain timestep.
> 4. In Figure 3 the blud rectangles denote surrounding vehicles and the ego-vehicle is the white rectangle. Figure 3(a) illustrates a scenario where the presence of the surrounding vehicles in the adjacent lane prevents lane changing. In this scenario, the TT planner tends to make overly aggressive maneuvers, resulting in collisions with the vehicles in the adjacent lane. On the other hand, the UNREST planner adopts a cautious planning approach and takes timely actions to stop before any collision occurs.
>
> Hope these clarifications could address your concerns and facilitate a more comprehensive understanding of the insights and contributions of this paper.
>
> ### References
> [1] Addressing Optimism Bias in Sequence Modeling for Reinforcement Learning, ICML 2022.
>
> [2] UMBRELLA: Uncertainty-Aware Model-Based Ofﬂine Reinforcement Learning Leveraging Planning, NeurIPS 2021.

---

> ### Author Response · Authors · 2023-11-19
> **To reviewer nCLY: looking forward to your reply.**
>
> Thank you for your valuable guidance in improving our paper (please see the updated pdf and our response with more discussion and experimental results). Here we humbly ask for your feedback at your convenience such that we may still have the chance to discuss with you and chance to improve our final version before the close of the rebuttal window.

---

> ### Author Response · Authors · 2023-11-21
> **To reviewer nCLY: looking forward to your reply.**
>
> Since the rebuttal period is almost over, we are wondering whether our responses have addressed your concerns properly. Your feedback will definitely help reach a more reasonable decision on our submission. Thank you!

---

> > ### Comment · Reviewer_nCLY · 2023-11-21
> > **Response to authors**
> >
> > I appreciate the efforts made by the authors in preparing the rebuttal. I have a follow-up question.
> >
> > As the authors say, the accumulated uncertainty exists in reward-to-go in DT and value function estimation in RL. What about the other frameworks? For the autonomous driving task using RL seems not a promising option. In general, I am still not convinced by the importance of accumulated uncertainty in autonomous driving tasks since it seems that we don’t need to care about it if we don’t use the RL method. Please correct me if I am wrong.

---

> ### Author Response · Authors · 2023-11-22
> **Response to reviewer nCLY.**
>
> Thank you for the feedback. First and foremost, it is imperative for us to highlight that the application of RL in the context of autonomous driving planning has been increasingly gaining popularity in recent times [3, 4], contrary to your assertion that it is not a promising option (as stated in the first paragraph of our paper). As claimed in [4], **RL enables the planners to learn the causal relationship of driving through rewards while IL simply copies experts' actions**. In contrast to the performance limitations imposed by expert-based approaches such as IL [2], RL holds the potential to achieve performance that surpasses that of human experts. That's also the reason why our paper and [5, 6] choose offline RL driving planners.
>
> Next, we want to explicate that uncertainty accumulation exists in both rule-based [1] and IL [2] planners as well. For rule-based planners like [1], they generally make decisions by solving an optimization problem over multiple timesteps (model predictive control) and safety constraints. As a result, the uncertainty of environment transitions accumulates with each timestep of the lookup process, necessitating the selection of the most cautious action to ensure safety. For IL planners like [2], due to the relatively narrow scope of expert data in terms of included states, it is common practice to simulate driving trajectories beyond the dataset in order to minimize out-of-distribution (OOD) states during actual testing. In this sense, the uncertainty accumulates over the rollout timesteps, thereby introducing additional trajectories that do not exist in the original dataset to assist in imitation learning.
>
> Finally, based on all of our responses, we would like to express a few points. First, we believe that the "reward-cost tradeoff" and the "scope of uncertainty accumulation" mentioned by you cannot be seriously considered as weaknesses of the model, but rather as potential concerns raised by you. These concerns have now been thoroughly discussed and addressed. Secondly, for the issues you pointed out regarding the presentation in the paper, we have made targeted modifications or highlighted aspects in the original text that you may not have initially noticed, which we believe will provide readers with a clearer understanding of the paper's intent. Noticing the positive feedback from the other two reviewers (xhtH and PUEU), and that reviewer mZKV only highlighted deficiencies in writing, we sincerely hope that you will reconsider your evaluation of the paper (since your concerns are now adequately addressed). Looking forward to your reply.
>
> [1] Comprehensive Reactive Safety: No Need For A Trajectory If You Have A Strategy, IROS 2022.
>
> [2] ChauﬀeurNet: Learning to Drive by Imitating the Best and Synthesizing the Worst, RSS 2018.
>
> [3] End-to-End Urban Driving by Imitating a Reinforcement Learning Coach, ICCV 2021.
>
> [4] DriveAdapter: Breaking the Coupling Barrier of Perception and Planning in End-to-End Autonomous Driving, ICCV 2023.
>
> [5] Addressing Optimism Bias in Sequence Modeling for Reinforcement Learning, ICML 2022.
>
> [6] UMBRELLA: Uncertainty-Aware Model-Based Ofﬂine Reinforcement Learning Leveraging Planning, NeurIPS 2021.

---

> > ### Comment · Reviewer_nCLY · 2023-11-22
> > **Response and clarification**
> >
> > Thanks to the authors for providing more evidence for their points. As an independent reviewer, my review is based on my personal experience and scientific sense. All other reviewers giving positive scores **DOES NOT** mean I should increase my score to match theirs. I can't decide the acceptance of this paper and I am just responsible for providing feedback from my perspective.

---

> > > ### Author Response · Authors · 2023-11-22
> > > **Response to reviewer nCLY.**
> > >
> > > Thank you for the feedback. We acknowledge your perspective in reviewing the paper from your own standpoint. However, we hope that you can also take into account the understanding of other reviewers which may complement yours. Regarding the concerns you raised earlier, we believe we have addressed them adequately. If there are any remaining issues, we would appreciate further amicable communication.

---

### Official Review · Reviewer_mZkV · 2023-11-01

**Soundness:** 3 good
**Presentation:** 2 fair
**Contribution:** 3 good
**Rating:** 5
**Confidence:** 3

**Summary:**

The paper proposes a method that adopting conditional mutual information of two network output distribution as the measure of uncertainty. And some numerical superiority is demonstrated.

**Strengths:**

uncertainty is an important topic in RL topics especially in autonomous driving. The general idea of separating environmental stochasticity versus action return is interesting.
Some theoretical justifications are provided to hint some insights of the method.
Numerical experiments show some superiority.

**Weaknesses:**

The paper is not well written.
1. A lot of notations are used before definition. For example, section 4.2 \tau^{ret}_{<t}
2. Some "terms" and "notation" are used without definition or even description, which makes very difficult to understand. For example, Figure 1(a) Task I and Task II (b) normalized ground Truth.
3. A lot of key contents description is very "verbal" and "handwaving", leaving huge ambiguity of the understanding of the algorithm.
4. If DT is a fundamental base of the method, having a brief introduction is more friendly to new readers.

**Questions:**

Does the "Transformer" term in section 4,2 means two independent network for x_{a_t} and x_{s_t}. Please give more concrete/clear description using notations flow of your method in general instead of long words description.

---

> ### Author Response · Authors · 2023-11-17
> **Response to Reviewer mZkV (1/1).**
>
> Thank you for the valuable advice on improving our presentation. We are pleased that you acknowledged the contribution and soundness of our paper. For your questions, we answer them one-by-one as follows (also see the revised paper in the updated pdf):
>
> > ***Q1: The paper is not well written. (1) Some terms and notations are used without definition or description. (2) Some key contents description is "verbal" and "handwaving", making it ambiguous to understand the algorithm. (3) A brief introduction of DT is needed in the paper. (4) Use more concrete/clear description by notations in the method instead of long words description.***
>
> **R1:** Thank you for pointing out the issues in our writing. We have made necessary modifications to the corresponding content in our updated version of paper (highlight in green and blue). Moving forward, here we provide individual responses to all of your questions.
>
> 1. Regarding the issue of notation, we can ensure that all symbols used are clearly defined in the paper. Like the example $\boldsymbol{\tau_{<t}^{ret}}$ you raised in Section 4.2, we have defined in Section 3.2 that $\mathbf{x_{<t}}$ denotes tokens from step 1 to $(t-1)$, and in Section 4.2 we define $\boldsymbol{\tau^{\text{ret}}}=\{(s_t,a_t)\}_{t=1}^T$ to be all state-action pairs involved in the driving trajectory. Besides, we have clarified Task I, Task II, and Normalized Ground Truth in Figure 1. Specifically, Figure 1(a) is composed of two driving tasks: go straight then turn right, and the x-axis in Figure 1(b) denotes the ground truth return in original driving trajectories.
> 2. For the issue of verbal expressions, since you haven't provided specific examples (**we would greatly appreciate it if you could respond with specific points in the paper that were unclear to you**, as this could really help in improving our writing), we have taken it upon ourselves to identify and modify certain aspects of the paper. For example, in the introduction, we further elaborate on the content depicted in Figure 1 and the insights derived from it. In addition, we have also provided a clearer introduction to our sequence segmentation and uncertainty-guided planning approach.
> 3. We acknowledge your point that since DT is a crucial foundation of this paper and warrants a separate section to provide an overview of its fundamental concepts and contents. Specifically, we have written in Section 3.2 of the original paper a description of the training and testing details for DTs, which you may miss in the first time of reading.
> 4. We apologize for any misunderstanding caused by the informal expression used in our formula. Typically, we use two separate transformers to get $\tilde{x_{s_t}}$ and $\tilde{x_{a_t}}$. And to express the formula more concisely, we have used the notation $\mathcal T$ to replace the long word 'Transformer' in Equation (3) and (7).
>
> Hope these improvements we have made in our writing could address your concerns and help you gain a clearer understanding of the insights and contributions of this paper.

---

> ### Author Response · Authors · 2023-11-19
> **To reviewer mZkV: looking forward to your reply.**
>
> Thank you for your valuable guidance in improving our writing (please see the updated pdf and our response with more discussion). Here we humbly ask for your feedback (since we find you overall like our work except the presentation) at your convenience such that we may still have the chance to discuss with you (e.g. pointing out the specific aspects within the text that you find to be 'verbal' or 'handwaving') and chance to improve our final version before the close of the rebuttal window.

---

> > ### Comment · Reviewer_mZkV · 2023-11-23
> >
> > Thanks for your work for paper revision. I changed presentation score to fair.

---

> ### Author Response · Authors · 2023-11-21
> **To reviewer mZkV: looking forward to your reply.**
>
> Since the rebuttal period is almost over, we are wondering whether our revised paper and responses have addressed your concerns properly. Your feedback will definitely help reach a more reasonable decision on our submission. Thank you!

---

### Official Review · Reviewer_PUEU · 2023-11-02

**Soundness:** 3 good
**Presentation:** 3 good
**Contribution:** 2 fair
**Rating:** 6
**Confidence:** 3

**Summary:**

This paper studies the problem of modeling driving as an *offline* stochastic sequence prediction problem. This is inspired by the recent success of approaches like the decision transformer. They note that for tasks like driving, existing approaches that just perform sequence prediction on an offline data set conflate the success a of a particular action with the stochastic transition necessary to replicate the success. This results in the accumulation of uncertainty in future predicted value, making existing approaches overly optimistic and unsuitable for such environments. The paper proposes explicitly modeling uncertainty in certain parts of the sequence prediction and *not* condition on the target reward in such cases. This naturally segments the problem into a sequence of tasks which as the paper points in natural in the driving context. The paper shows with several studies showing the efficacy of their approach.

**Strengths:**

1. The paper addresses some of the emerging concerns with naively applying sequence prediction systems such as the decision transformer to decision making tasks. Namely, that the underlying domain can be stochastic.

2. The approach is straightforward and motivates the way it builds off of existing work.

3. The method is empirically demonstrated to perform well against existing baselines including  other approaches which attempt to model uncertainty.

4. The ablation and experiment design seem sufficiently well crafted and the approach is well motivated enough that I generally believe the key results.

**Weaknesses:**

1. The exposition, particularly the first half of page 2 is hard to understand.
    a. For example, its not explained what a highly representative environment model is and why it might not be learnable. I understand this is related work, but the reader is completely left in the draft about why the presented approach avoids this.
    b.  Task I+II is initially presented in a manner that is unclear that + here means sequential composition, e.g. perform Task I and then Task II. This makes sense once finishing the page and the segmentation, but the exposition seems like it could be improved significantly.

2. The uncertainty threshold seems like a key hyperparameter and it seems unclear how to set it. Further, it seems that mainly driving tasks have large regions of uncertainty that should themselves be segmented, e.g., a sequences of nearby intersections. How to generalize the segmentation approach to this setting is unclear.

3. While there's ample empirical evidence for this approach's success, there isn't a clear theoretical framework to connect it to. I see a tentative connective with work on causal vs non-causal entropy regularization and it's similar effect on creating over optimistic agents. A deeper theoretical basis would be appreciated.

**Questions:**

The experiments and setup seem to be for the fully observable context, save for the other driver's implicit latent states.

That said, in the automotive setting a non-trivial amount of uncertainty is derived from the partial observability of the environment, e.g., where exactly is the crossing region in an intersect, what is the drivable area, what are that boundaries, where are the obscured pedestrians and bikes, etc.

How could the presented approach be adapted to such settings?

---

> ### Author Response · Authors · 2023-11-17
> **Response to Reviewer PUEU (1/3).**
>
> Thank you for the positive feedback and constructive suggestions that can help us to improve the paper. We are happy that you acknowledge our work’s motivation, technical quality and presentation. For your questions, we provide additional experimental results and discussions as follows (also see the revised paper in the updated pdf):
>
> > ***Q1: Some parts of the introduction is somewhat difficult to understand.***
>
> **R1:** Thank you for pointing out the issue with our unclear statement. We have modified the introduction in the newly updated version (highlighted in green and blue) and here we answer your problems respectively.
> 1. About the 'highly representative environment model', it usually means the state transition function $P(s'|s,a):\mathcal{S}\times\mathcal{A}\times\mathcal{S}\rightarrow[0,1]$ in reinforcement learning. Previous methods [2, 3, 4] attempt to fit the state transition model to deal with transition uncertainty. However, while adding complexity, these methods are applicable only when state transition functions can be learned adequately, which is often not the case for driving because of the uncertainty brought by complex interactions and partial observability. Our method bypasses the challenging high-dimensional state transition function learning by computing uncertainty through conditional mutual information $u_t=D_{\text{KL}}\big(p_{\varphi_a}(R_{t}|\boldsymbol{\tau_{<t}^{\text{ret}}}),p_{\varphi_s}(R_{t}|\boldsymbol{\tau_{<t}^{\text{ret}}},s_{t})\big)$, where the return distributions can be effectively learned with fewer resources.
> 2. Regarding Figure 1(a) and its corresponding explanation, it represents a driving scenario composed of two tasks (now denoted as Task I & II), starting with going straight (Task I) and followed by a right turn (Task II). In this scenario, the uncertainty in the environment is relatively low during the straight driving process, while the variance in the distribution of rewards increases substantially considering Task I & II as a whole due to the uncertain behavior of other vehicles in the environment during the right turn. Therefore, we propose to condition solely on the truncated return from Task I (going straight) rather than Task I & II to mitigate the influence of environmental stochasticity (less stochastic timesteps, lower return variance) for DT training, which still preserves rewards over sufficient timesteps for the optimization of current actions.
>
> Hope these modifications in our presentation can help you better understand the main insights and contributions of our work.

---

> ### Author Response · Authors · 2023-11-17
> **Response to Reviewer PUEU (2/3).**
>
> > ***Q2: Missing discussion about how to set the uncertainty threshold in experiments. Besides, how to generalize the segmentation approach to scenarios like sequences of nearby intersections.***
>
> **R2:** Thank you for pointing out this important problem of UNREST's practical usage. Due to space constraints, **we have postponed the discussion on the selection of the uncertainty threshold to Appendix F**, which you may have overlooked during initial reading. Specifically, in Figure 5(a), we plot the histogram of uncertainty value. As we can see the majority of uncertainty estimation associated with state transitions falls within the range of $0$ to $10^{-4}$, aligning with intuitive expectations. In the context of autonomous driving environments with sparse vehicle presence and a limited number of traffic signals, the stochasticity of environment transitions tends to be relatively low. The uncertainty threshold $\epsilon$ is accordingly chosen as 3.0 to cover corner cases with large uncertainties.
>
> Second, to generalize our segmentation mechanism to driving tasks which have large regions of uncertainty that should themselves be segmented, you can refer to results in Figure 7. **By adjusting the uncertainty threshold $\epsilon$ and the minimum sequence length $c$, we can flexibly control the distribution of the segmented sequences.** For your case, we can simply decrease the values of $\epsilon$ and $c$ (as shown in the third subplot of Figure 7). By using a smaller value of $c$, each uncertain segment can promptly stop when encountering consecutive certain states. Similarly, a smaller value of $\epsilon$ can detect and initiate new uncertain segments more quickly when entering a new uncertain region. So it appropriately helps for further segmentation within large uncertain areas.
>
>
> > ***Q3: Though with ample empirical evidence for UNREST's success, there isn't a clear theoretical framework to connect it to.***
>
> **R3:** We thank the reviewers for pointing out our oversight in the paper. In fact we have detailedly discuss UNREST's theoretical generalizability in our original paper, as you can find in Proposition 1. Due to the page limit, the proposition in the main context is explained in an informal manner, with the formal statement and proof left to Appendix B. Specifically, the proposition is inspired by [1] and reveals that **under natural assumptions in driving environments, UNREST can generalize to achieve high returns with bounded error at less stochastic states if expert demonstrations cover corresponding actions.** The relevant content has been highlighted in the new version of our paper, where you can find further details for discussion. Besides, I find the causal/non-causal entropy regularization mentioned by you to be an exceptionally intriguing aspect. It is possible that we can explore this point further to explain how UNREST addresses the issue of overly-optimistic behavior of agents.

---

> ### Author Response · Authors · 2023-11-17
> **Response to Reviewer PUEU (3/3).**
>
> > ***Q4: How could UNREST generalize to partially observable environments where the drivable area, boundaries, pedestrains and bikes are not fully observed?***
>
> **R4:** Thank you for highlighting this interesting question regarding our experimental design. As you aptly pointed out, the uncertainty in the driving environment can arise from both the latent states of surrounding vehicles and partial observability. In this response, I would like to elucidate the capability of UNREST in effectively handling the uncertainty introduced by partial observability.
>
> **Theoretically,** we compute the uncertainty by the conditional mutual information $u_t=D_{\text{KL}}\big(p_{\varphi_a}(R_{t}|\boldsymbol{\tau_{<t}^{\text{ret}}},p_{\varphi_s}(R_{t}|\boldsymbol{\tau_{<t}^{\text{ret}}},s_{t})\big)$. So regarding the uncertainty introduced by partial observability, such as pedestrians who were obstructed and not visible in the previous step but are now visible, can result in a significant change in return distribution $p_{\varphi_s}(R_{t}|\boldsymbol{\tau_{<t}^{\text{ret}}},s_{t})$ from $p_{\varphi_a}(R_{t}|\boldsymbol{\tau_{<t}^{\text{ret}}})$ because of the newly captured information.
>
> **From the experimental perspective,** we have discussed in Appendix D in the paper about our state space, which is similar to that in [2]. Specifically, the ego vehicle can only obtain distances of nearby vehicles within its limited field of view and perceive the boundaries of a limited portion of the target route, which is exactly consistent with your proposed partially observable setting. Building upon this, our experimental results further demonstrate the superior ability of UNREST compared to previous baselines in addressing the uncertainty brought by partial observability. Besides, we also visualize in Figure 4.(b) about the uncertainty curve when the ego-vehicle is passing the green traffic light. As you can see, while approaching the green light, the partial observability about the change of light state causes the uncertainty to rise quickly. But after the vehicle has moved away from the traffic light, the uncertainty immediately drops back below the threshold. This interpretable result also validate the effectiveness of our uncertainty measure.
>
> ### References
> [1] When does return-conditioned supervised learning work for ofﬂine reinforcement learning? NeurIPS 2022.
>
> [2] Addressing Optimism Bias in Sequence Modeling for Reinforcement Learning, ICML 2022.
>
> [3] You Can’t Count on Luck: Why Decision Transformers and RvS Fail in Stochastic Environments, NeurIPS 2022.
>
> [4] Dichotomy of control: Separating what you can control from what you cannot, ICLR 2022.

---

> ### Author Response · Authors · 2023-11-21
> **To reviewer PUEU: looking forward to your reply.**
>
> We express our sincere gratitude for your valuable guidance in improving our paper (please see the updated pdf and our response with more discussion). Here we humbly ask for your feedback at your convenience such that we may still have the chance to discuss with you and chance to improve our final version before the close of the rebuttal window.

---

### Official Review · Reviewer_xhtH · 2023-11-04

**Soundness:** 2 fair
**Presentation:** 2 fair
**Contribution:** 3 good
**Rating:** 6
**Confidence:** 2

**Summary:**

The paper addresses a major challenge of "return-conditioned supervised learning" methods like Decision Transformers (DTs) in stochastic environments. DTs can be overly optimistic in stochastic environments when some of these expert (high reward) trajectories arise due to accidental environment transitions. The authors propose a "practical" solution for this in the self-driving domain, using 'uncertainty accumulation' and 'temporal locality' properties specific to this domain. They segment the trajectories into deterministic/certain (low uncertainty) and stochastic/uncertain (high uncertainty) regions using information theoretic metrics. In deterministic regions, they use a "truncated return conditioned DT", while in the stochastic regions, they allow the policy to cautiously follow the expert without any return conditioning. Empirical results demonstrate UNREST's superior performance in various driving scenarios compared to the state-of-the-art offline RL methods.

**Strengths:**

These are the strengths in my opinion:

1) Studies and proposes a practical solution to a major drawback of "return-conditioned supervised learning" methods like DTs in stochastic environments (specifically self-driving domain).
2) The segmentation based on uncertainty and associated decision-making intuitively makes sense and has not been explored before in the context of DTs, Upside Down RL [1,2] etc in stochastic environments. to the best of my knowledge.
3) The results on the CARLA simulator are strong compared to the baselines.

**Weaknesses:**

These are the weaknesses in my opinion:

1) Evaluated only on a single environment.
2) The idea of using truncated returns is not entirely novel though and has been used in the upside-down RL methods [1,2].
3) The algorithm assumes the data have to come from an expert if I understand correctly (especially for the logic to work in uncertain regions). The authors use expert data (from the autopilot) in the training process. Isn't this a drawback when compared to existing offline RL baselines which can also leverage suboptimal trajectories/data?

**Questions:**

1) Would this method scale to other stochastic environments like the ones used in baseline algorithms [3] and [4] ?
2) Would this method be applicable if the offline dataset contains suboptimal trajectories ??
3) Is there a better strategy the method can follow in "uncertain" regions than just following the expert? Like somehow being able to use the expected Returns or something on that line.


**References**
[1] Schmidhuber, Juergen. "Reinforcement Learning Upside Down: Don't Predict Rewards--Just Map Them to Actions." arXiv preprint arXiv:1912.02875 (2019).

[2] Srivastava, Rupesh Kumar, et al. "Training agents using upside-down reinforcement learning." arXiv preprint arXiv:1912.02877 (2019).

[3] Sherry Yang, Dale Schuurmans, Pieter Abbeel, and Ofir Nachum. Dichotomy of control: Separating what you can control from what you cannot. In The Eleventh International Conference on Learning Representations, 2022.

[4] Keiran Paster, Sheila McIlraith, and Jimmy Ba. You can’t count on luck: Why decision transformers fail in stochastic environments. Advances in neural information processing systems, 2022.

---

> ### Author Response · Authors · 2023-11-17
> **Response to Reviewer xhtH (1/2).**
>
> Thanks for your time and constructive feedback. We are pleased that you acknowledged the contribution and experiments of our work. Here we provide more experimental results and insights to address your concerns (also see the revised paper in the updated pdf):
>
> > ***Q1: UNREST is only evaluated in a single environment. Would it scale to other stochastic environments like those in related works?***
>
> **R1:** Thank you for pointing out the problem of our experiments. The reason why we only study the driving environment is that, though as a general offline RL algorithm that can be applied to any RL environments, **UNREST is specially designed to address two major challenges in autonomous driving: long-horizon planning and environmental stochasticity.** To make the segmentation strategy proposed by UNREST effective, two essential factors are required：1）'uncertainty accumulation' property that ensures the truncated return is less affected by the environments than the global return, to provide effective supervison for policy learning; 2) 'temporal locality' property that makes optimizing future returns over a sufficiently long horizon similar to the optimization of global return. We have identified these two properties in the context of autonomous driving, but they may not necessarily apply in other environments. **We have also tested UNREST on the conventional D4RL benchmark (approximately deterministic environments, results in Appendix F.5)**, where it just attains similar results to vanilla DTs. Similarly, SPLT [3] is also only tested on the CARLA simulator. Therefore, we believe that the autonomous driving environment is the most suitable environment to test our proposed algorithm.
>   | Environment | DT    | TT      | SPLT  | ESPER  | UNREST   |
>   | ------- | --------------------------- | --------------------------- | --------------------------- | --------------------------- | ----------------------------- |
>   | HalfCheetah-Stochastic  | $75.8\pm1.2$ | $83.7\pm2.8$ | $88.6\pm2.2$ | $77.4\pm5.6$ | $\mathbf{90.5\pm3.2}$ |
>   | Hopper-Stochastic     | $89.4\pm3.3$ | $93.4\pm3.6$ | $94.8\pm2.7$ | $92.5\pm6.9$ | $\mathbf{97.4\pm4.4}$ |
>
> In the above table, **we test UNREST in the customized stochastic D4RL environments from DoC [4],** which add time-correlated Gaussian noise to actions. The results are also added in Appendix F.5, where UNREST achieves highest return among the models.
>
> > ***Q2: The idea of using truncated return is not entirely novel enough.***
>
> **R2:** We appreciate reviewer's attention to the novelty of our work. Here we need to clarify one significant point that the key novelty of UNREST lies in **the low-resource occupancy uncertainty estimation method, along with the proposed sequence segmentation and designs of conditions based on it.**
> - **Novelty 1:** Different from previous works [2, 3, 4] that attempt to fit high-dimensional state transition functions and introduce complex VAE or adversarial training, **we introduce for the first time the utilization of conditional mutual information to estimate uncertainty**. As discussed in the paper, the complex interactions, partial observability, and long trajectories challenges training of previous works, where our proposed mutual information can be effectively learned with fewer resources, as validated in experiments.
> - **Novelty 2:** Back to the fundamental objective of this paper, we propose **a novel sequence segmentation and designs of conditions for customizing DTs in stochastic driving environments**. Specifically, we segment sequences into certain and uncertain parts according to our uncertainty measurements, and learn to behave aggressively and cautiously within them, respectively.
> - **Novelty 3:** We also **summarize two essential properties of driving environments**: 'temporal locality' and 'uncertainty accumulation' properties, based on which we **theoretically prove** that replacing conditioned returns in original DT with truncated returns that only consider returns to the next segmented positions can help UNREST generalize to attain high rewards.
>
> Therefore, while it is true that truncated return has been employed in previous works [1], in this paper based on our discovered properties, we introduce a novel specific segmentation method based on our proposed uncertainty estimation techiniques and provide corresponding theoretical justifications. This represents a distinct contribution that was absent in prior literature.
>
> Hope this clarification will assist you in better understanding the key insights and novelty of this paper.

---

> ### Author Response · Authors · 2023-11-17
> **Response to Reviewer xhtH (2/2).**
>
> > ***Q3: The algorithm assumes expert-level data to function in uncertain regions. Would it be applicable in offline datasets with suboptimal trajectories? Is there a better strategy the method can follow in uncertain regions than just following the expert?***
>
> **R3:** Thank you for highlighting this crucial issue regarding the scope of applicability of UNREST. In fact, the decision to resort to direct behavior cloning (BC) in uncertain parts within UNREST can be attributed to the compromise made due to the unreliability of returns, which hinders the provision of effective supervision. We opted for this approach for two reasons: (1) Firstly, **BC is a lightweight and easily implementable algorithm** that simply requires setting the original conditioned return as a dummy token. (2) Secondly, **expert data collection is relatively easy in the context of autonomous driving environments**. However, it should be noted that the AutoPilot provided by CARLA has multiple fail cases and cannot be considered an optimal expert.
>
> In order to propose a strategy that can be employed in uncertain regions when there exists suboptimal trajectories, we propose a simple yet effective UNREST variant that **additionally fit a state value function**. The remaining logic is consistent with MARWIL [5], where we utilize neural network estimation of advantages to reweight actions in the BC objective. In this way, UNREST can learn to identify and choose to imitate actions in optimal trajectories. The results are recorded in the following table.
>   | Planner | Driving Score $\uparrow$    | Success Rate$\uparrow$      | Route Completion$\uparrow$  | Infraction Score$\uparrow$  | Normalized Reward$\uparrow$   |
>   | ------- | --------------------------- | --------------------------- | --------------------------- | --------------------------- | ----------------------------- |
>   | Reweighted BC  | $64.2\pm2.4$ ($63.8\pm2.3$) | $57.5\pm3.3$ ($59.3\pm4.7$) | $85.5\pm2.7$ ($91.0\pm5.0$) | $71.4\pm2.5$ ($65.1\pm3.7$) | $0.68\pm0.03$ ($0.67\pm0.02$) |
>   | Original Model     | $63.5\pm3.2$ ($62.9\pm4.0$) | $54.5\pm7.0$ ($57.5\pm5.4$) | $83.8\pm3.1$ ($90.0\pm6.0$) | $70.2\pm2.8$ ($62.9\pm3.8$) | $0.64\pm0.04$ ($0.65\pm0.03$) |
>
> Since the offline data collected by AutoPilot can be suboptimal, we find reweighted BC can effectively down-weight those suboptimal actions and improve UNREST's driving scores at both training and new scenarios. However, due to its significant increase in model complexity, we still recommend the original UNREST as the base planner.
>
> ### References
> [1] Reinforcement Learning Upside Down: Don't Predict Rewards--Just Map Them to Actions, Arxiv 2019.
>
> [2] You Can’t Count on Luck: Why Decision Transformers and RvS Fail in Stochastic Environments, NeurIPS 2022.
>
> [3] Addressing Optimism Bias in Sequence Modeling for Reinforcement Learning, ICML 2022.
>
> [4] Dichotomy of control: Separating what you can control from what you cannot, ICLR 2022.
>
> [5] Exponentially Weighted Imitation Learning for Batched Historical Data, NeurIPS 2018.

---

> > ### Comment · Reviewer_xhtH · 2023-11-19
> >
> > Thank you for your reply... Could you please detail how exactly was the custom stochastic D4RL created? Did you use some open-source data with stochasticity from previous works or generated on your own data?

---

> ### Author Response · Authors · 2023-11-19
> **Response to Reviewer xhtH.**
>
> Thank you for the feedback. For your question, as we have stated in the previous response, we simply adopt the same environment setting as [4]. While the original D4RL environments are deterministic by default, we modify them by introducing time-correlated Gaussian noise to the actions before inputing the action into the physics simulator during data collection and evaluation. We find these simple modifications to environments greatly harm DT's performance, where UNREST succeeds in obtaining performance improvements.

---

> > ### Comment · Reviewer_xhtH · 2023-11-21
> >
> > Thank you...  the additional experiments are appreciated... Since you wanted to replicate the same exact setup as in DoC [4] , why didn't you choose the same environment as them (AntMaze) ? This would have been more convincing and a fairer comparison, since you can directly take the baseline results from their Figure 4... This also would have convinced the reviewers that the baselines were properly tuned etc

---

> > > ### Author Response · Authors · 2023-11-21
> > > **Response to Reviewer xhtH.**
> > >
> > > Thank you for your reply. As you can see from Figure 4 in DoC, the environments Hopper and HalfCheetah were all evaluated in their paper, for which we choose them as new testing environments. Regarding why we did not choose the AntMaze environment, it is because their approach to introducing uncertainty in the environment, as described in their paper, is relatively simplistic compared to Hopper and HalfCheetah's method (adding Gaussian noise to the reward function). We believe that results of Hopper and HalfCheetach are more persuasive and compelling.

---

> > > > ### Comment · Reviewer_xhtH · 2023-11-22
> > > >
> > > > Thanks for your reply!!
> > > >
> > > > The DoC paper did not modify the D4RL agents for Half Cheetah or Hopper as in your case... They used the D4RL offline RL benchmark only for the ant maze (ant-maze-v0 and ant-umaze-diverse-v0) environments... Whenever they used a pre-collected deterministic dataset from D4RL, they used a different way to create stochasticity by modifying rewards... Why do you think it's simplistic? Were the baselines including DoC performing equally well when you tried this out?
> > > >
> > > > A fairer, easier and more trustworthy comparison would have been using the same exact data setup (ant-maze-v0 and ant-umaze-diverse-v0) and just reporting the numbers given by the DoC paper for the baselines... This would be been more compelling to me over a self-created baseline and evaluation based on that...

---

> > > > > ### Author Response · Authors · 2023-11-23
> > > > > **Response to Reviewer xhtH.**
> > > > >
> > > > > Thanks for your reply!
> > > > >
> > > > > There might be some misunderstandings. To clarify, you can refer to the original paper of DoC [4]. In order to locate the relevant information, please refer to Chapter 6.3 and Figure 4. By doing so, you will discover that apart from the AntMaze environment, the authors of the paper also conducted experiments on environments such as Hopper and HalfCheetah within the MuJoCo framework (which is also collected in the D4RL benchmark). For these two environments,  we adopt exactly the same approach as DoC to introduce uncertainty by introducing time-correlated Gaussian noise to the actions before inputting the action into the physics simulator, which is more complex than simply modifying the reward functions.
> > > > >
> > > > > Regarding the adoption of the results from DoC mentioned in your statement, due to the unavailability of the source dataset, we can only reproduce the results through the means described in the original text, thereby demonstrating the effectiveness of UNREST.

---

### Author Response · Authors · 2023-11-17
**General Response**

Dear Area Chairs and Reviewers,

We sincerely thank the reviewers for their time, thorough insightful reviews and constructive suggestions. Overall, it is heartening to note that most of the reviewers found our work to be well motivated(xhtH, PUEU, mZkV, nCLY), novel (xhtH, PUEU, mZkV), and experimental solid (PUEU, mZkV, nCLY). To clarify some potential misunderstandings of our paper, we first address some shared concerns of reviewers:

- **UNREST's scope of applicability:** (1) Notably, though specially designed for driving environments, **UNREST can also be used in other stochastic environments** like the ones studied in [3]. The results are recorded in Appendix Table 10 in the revised paper, where UNREST achieves the highest return among the models. (2) Besides, **UNREST can also be generalized to datasets with suboptimal trajectories**. In this case, we need to modify the policy UNREST follows in uncertain parts. Specifically, we propose to additionally fit a state value function and use the advantage estimation from it to reweight actions in the dataset, like that in MARWIL [1]. Results are recorded in Table 8 and 9 in the revised paper, where we find reweighted BC can effectively down-weight those suboptimal actions and improve UNREST's driving scores at both training and new scenarios. However, due to its significant increase in model complexity, we still recommend the original UNREST as the base planner. (3) **Our uncertainty estimation is also applicable in RL frameworks other than DT**. For example, [2, 4] propose to learn transition models for monte-carlo simulation (one fundamental method in RL). Due to the significant bias associated with single-step rollouts, they have all opted for multi-step beam search to find the most effective and safe actions. By combining our uncertainty estimation method, we can effectively identify specific uncertain states that require branching and rollout. This can greatly help reduce the number of simulations and improves planning efficiency.
- **Presentation clarity:** We thank the reviewers' feedback regarding the issues in our writing. We have addressed these concerns in the revised version of the manuscript by making corrections to the figures, notations, and explanations. For example, regarding Figure 1(a) and its corresponding explanation, it represents a driving scenario composed of two tasks (now denoted as Task I & II), starting with going straight (Task I) and followed by a right turn (Task II). In this scenario, the uncertainty in the environment is relatively low during the straight driving process, while the variance in the distribution of rewards increases substantially considering Task I & II as a whole due to the uncertain behavior of other vehicles in the environment during the right turn. Therefore, we propose to condition solely on the truncated return from Task I (going straight) rather than Task I & II to mitigate the influence of environmental stochasticity (less stochastic timesteps, lower return variance) for DT training, which still preserves rewards over sufficient timesteps for the optimization of current actions. In addition, we have also provided a clearer introduction to our sequence segmentation and uncertainty-guided planning approach.

We provide extraordinary experiment results and detailed answers to all the questions raised by the reviewers in the following individual responses. Besides, we have also revised the paper w.r.t. the suggestions of the reviewers, which are highlighted in green (key contents that some reviewers missed in initial reading) and blue (revised content).


### References
[1] Exponentially Weighted Imitation Learning for Batched Historical Data, NeurIPS 2018.

[2] Addressing Optimism Bias in Sequence Modeling for Reinforcement Learning, ICML 2022.

[3] Dichotomy of control: Separating what you can control from what you cannot, ICLR 2022.

[4] UMBRELLA: Uncertainty-Aware Model-Based Ofﬂine Reinforcement Learning Leveraging Planning, NeurIPS 2021.

---

### Author Response · Authors · 2023-11-19
**Inquiry for post-rebuttal comments.**

Dear Reviewers:

Thank you again for your wisdom and valuable comments. We have provided experimental or complete explanations for all the questions. Since the discussion period is approaching its end, we would be glad to hear from you whether our rebuttal has addressed your concerns. Feel free to comment on our rebuttal if you have further questions and considerations.

---

### Meta-Review · Area_Chair_LAbn · 2023-12-16

**Metareview:**

The paper proposes a new algorithm for offline policy optimization in stochastic environments. The algorithm relies on segmenting trajectories into low uncertainty and high uncertainty regions based on the entropy in the transition dynamics. The empirical validation is done on the Carla environment. The reviewers appreciated the importance of the problem and acknowledged that the method was novel relative to prior works and had good empirical results on the self-driving domains. The major concerns with this work were regarding presentation, over-claiming of scope, and the restrictive assumptions under which experiments were designed and compared with relevant baselines. Some of these were partially addressed during the rebuttal process. Yet, there are some major points that still stick out for the reviewers. One is the assumption of doing direct BC for low uncertainty regions -- as reviewers noted, this hardly makes sense for offline RL where policies can be suboptimal. The authors' suggested fix during the rebuttal needs a more thorough examination across multiple environments. Relatedly, the work could do a better job acknowledging the differences with prior work (e.g., DOC) empirically. The reviewers had concerns regarding the selective experiments in the paper even after considering the experiments in the appendix (e.g., Tab 10 which seems to have much fewer environments than DoC which is not even directly compared to despite being a very relevant baseline). Overall, the work needs significant contextualizing and empirical validation to justify the proposed approach.

**Justification For Why Not Higher Score:**

Conceptual flaws in the algorithmic design/framing of problem as offline RL. Limited empirical evaluation.

**Justification For Why Not Lower Score:**

N/A

---

### Decision · Program_Chairs · 2024-01-16

Reject